# AUToCATE: END-TO-END, AUTOMATED TREATMENT EFFECT ESTIMATION

## ABSTRACT

Accurate estimation of heterogeneous treatment effects is critical in domains such as healthcare, economics, and education. While machine learning (ML) has led to significant advances in estimating conditional average treatment effects (CATE), real-world adoption of these methods remains limited due to the complexity of implementing, tuning, and validating them. To this end, we advocate for a more holistic view on the development of ML pipelines for CATE estimation through automated, end-to-end protocols. We formalize the search for an optimal pipeline as a counterfactual Combined Algorithm Selection and Hyperparameter optimization (CASH) problem. We introduce `AutoCATE`, the first automated solution tailored for CATE estimation that addresses this problem based on protocols for evaluation, estimation, and ensembling. Our experiments show how `AutoCATE` allows for comparing different protocols, with the final configuration outperforming common strategies. We provide `AutoCATE` as an open-source software package to help practitioners and researchers develop ML pipelines for CATE estimation.

## 1 INTRODUCTION

Accurately estimating causal effects is crucial for high-stakes decisions in domains such as healthcare, education, and economics. Despite advances in machine learning (ML) for estimating the conditional average treatment effect (CATE), real-world adoption remains limited due to the *complexity of developing ML pipelines for CATE estimation*. Methods often involve numerous hyperparameters, and their performance varies significantly across data sets and applications. Moreover, validating counterfactual predictions and tuning pipelines is highly challenging, and the performance of different evaluation criteria varies with the data generating process (Curth & van der Schaar, 2023). For practitioners unfamiliar with ML, such as clinicians or marketers, these challenges often outweigh potential benefits, hindering the practical use of these techniques. To overcome this, we advocate for *automated, end-to-end solutions* for learning ML pipelines for CATE estimation.

**The challenge of automated CATE estimation.** Despite automated ML (AutoML) making significant progress (see He et al., 2021), existing solutions do not address the unique challenges of CATE estimation. A key problem is the *lack of ground truth* CATE: the treatment effect is the difference between the outcomes with and without treatment, but only one of these outcomes is observed for each instance. Additionally, which outcome is observed depends on *confounding* variables (e.g., older patients may be more likely to receive treatment), leading to covariate shift (Shalit et al., 2017). Finally, CATE estimation pipelines are *more complex* than those in supervised learning. Metalearners combine multiple baselearners, possibly including both classification and regression models. Risk measures themselves also require predictions and, therefore, tuning of ML pipelines. These unique challenges complicate both the training and validation of ML pipelines and highlight the need for automated, end-to-end approaches tailored to CATE estimation, which is the focus of this work.

**Contributions.** To tackle these challenges, we propose a practical and comprehensive solution as the *automated, end-to-end construction and validation of ML pipelines* for CATE estimation:

• COUNTERFACTUAL CASH—We formalize the optimization of CATE estimation pipelines as a counterfactual Combined Algorithm Selection and Hyperparameter optimization (CASH) problem. Our solution, `AutoCATE`, automates the search for optimal configurations across preprocessors, metalearners, evaluators, baselearners, and their hyperparameters. The process is organized into three stages–*evaluation*, *estimation*, and *ensembling*–each including several design choices.

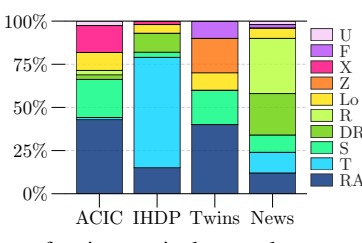

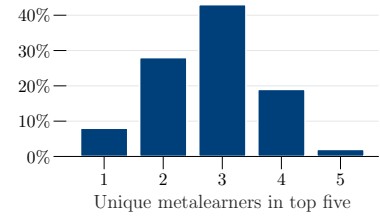

(a) How often is a particular metalearner optimal?

(b) How diverse are the best five metalearners? (IHDP)

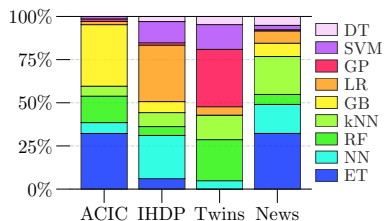

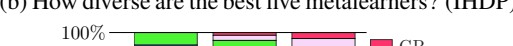

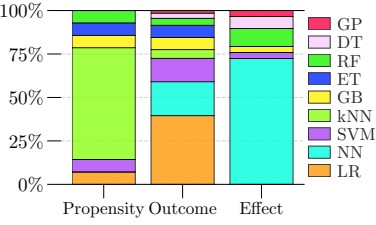

(c) How often is a particular baselearner chosen?

(d) What baselearner is best per model type? (IHDP)

Figure 1: **AutoCATE enables insights into CATE estimation.** We analyze hundreds of pipelines optimized by AutoCATE (see Section 5). *Metalearners*—(a) Different metalearners can be optimal for a data set, highlighting the need for searching across them. (b) The top five pipelines often feature a mix of different metalearners (e.g. $\{T, T, RA, RA, DR\}$: 3 unique types), showing that different metalearners can perform well and suggesting potential for combining them. *Baselearners*—(c) The chosen baselearners are also diverse, and (d) different model types favor different ones. Using a single baselearner is thus likely suboptimal, supporting our choice to tune submodels independently.

• END-TO-END PROTOCOLS—We develop end-to-end protocols that ensure robust performance across diverse data sets and applications. Our approach addresses key aspects often overlooked in CATE estimation, such as preprocessing, feature selection, or ensembling. This perspective uncovers novel *insights* (see Figure 1), *questions* (e.g., the intricate trade-off between using data for training or validation) and *solutions* (e.g., multi-objective optimization with different evaluation criteria).

• SOFTWARE PACKAGE—We provide AutoCATE as an open-source software package, enabling automated CATE estimation in a few lines of code. This way, we democratize access to advanced ML techniques for CATE estimation and make them accessible for practitioners unfamiliar with ML. Additionally, AutoCATE provides a platform for future research, encouraging research on all aspects of the ML pipeline for CATE estimation that supports practical, real-world applications.

## 2 RELATED WORK

Our work is most related to two areas in ML: (1) AutoML, and (2) CATE estimation and validation.

### 2.1 AUTOMATED MACHINE LEARNING (AUTOML)

AutoML focuses on the automatic and efficient construction of high-performing ML pipelines. This entails making a series of design choices regarding preprocessing, feature transformation and selection, ML algorithms, and hyperparameter tuning (Karmaker et al., 2021). As the optimal choices depend on the data and task, AutoML is essentially a *search problem*. While combinations could be tried randomly, more efficient search methods have been developed, e.g., based on Bayesian optimization (Bergstra et al., 2011; Snoek et al., 2012; Alaa & van der Schaar, 2018). Similarly, meta-learning has been applied to integrate information across other data sets in the search (Feurer et al., 2015). AutoML has made significant progress across data modalities, such as structured data (Erickson et al., 2020), text (Shi et al., 2021) or images (Bisong & Bisong, 2019). A critical aspect of AutoML is its accessibility, often provided through low-code solutions for practitioners unfamiliar with ML (LeDell & Poirier, 2020; Erickson et al., 2020; Jarrett et al., 2021; Wang et al., 2021).

Automated solutions exist for a wide range of tasks, including semantic segmentation (Chen et al., 2018), machine translation (So et al., 2019), reinforcement learning (Runge et al., 2019), or time

series forecasting (Jarrett et al., 2021). For more comprehensive overviews, see Elsken et al. (2019) and He et al. (2021). However, to the best of our knowledge, *AutoML has not yet been applied to CATE estimation*. As discussed, estimating treatment effects presents *unique challenges*, such as the absence of a ground truth, covariate shift due to confounding, and the need for intermediary models in metalearners and risk measures. These complexities render standard AutoML approaches ill-suited for CATE estimation and illustrate the need for approaches specialized to CATE estimation.

> **Research gap**—No existing AutoML solutions tackle the unique challenges of CATE estimation.

## 2.2 TREATMENT EFFECT ESTIMATION AND MODEL VALIDATION

**Estimation.** Various ML methodologies have been proposed for estimating treatment effects. Metalearners are general strategies for using standard supervised learning algorithms for CATE estimation (Künzel et al., 2019). Additionally, various ML algorithms have been adapted for CATE estimation, such as Gaussian processes (Alaa & van der Schaar, 2017), neural networks (Shalit et al., 2017; Yoon et al., 2018), decision trees (Rzepakowski & Jaroszewicz, 2012), or random forests (Wager & Athey, 2018; Oprescu et al., 2019). Notably, other parts of the ML pipeline are also more complicated when estimating treatment effects, such as missing value imputation (Berrevoets et al., 2023), feature selection (Zhao et al., 2022), and ensemble selection (Mahajan et al., 2023).

Building an ML pipeline for CATE estimation presents significant challenges, related to the absence of ground truth CATE and the number of design choices involved. Due to the *no free lunch theorem*, no ML algorithm be optimal in all possible settings. Additionally, there is no globally optimal metalearner, as performance similarly depends on the (unknown) data generating process and sample size (Curth & van der Schaar, 2021). Finally, *tuning is more involved*: for example, a *DR*-Learner combines four models (to estimate the propensity, the outcome per treatment group, and the final treatment effect)–each of which can be a different baselearner with separate hyperparameters.

**Model validation.** As the CATE is unobserved, various evaluation criteria have been proposed for validating CATE estimators. A common approach is the error in predicting the observed potential outcome $\mu$, i.e., the $\mu$-risk. However, this criterion has several limitations (Curth & van der Schaar, 2023; Doutreligne & Varoquaux, 2023): it does not account for confounding, may not accurately predict CATE error[1], and is not applicable to estimators that directly predict the CATE. To mitigate the first issue, an inverse propensity weighted variant $\mu_{\text{IPW}}$-risk, can be considered. Other evaluation criteria address all issues by constructing labels based on plug-in estimates (e.g., $S$- or $T$-risk) or metalearner pseudo-outcomes (e.g., $R$- and $DR$-risk), see Appendix B.2 for a detailed overview.

There is *no consensus on the optimal validation criterion*. While Schuler et al. (2018) and Doutreligne & Varoquaux (2023) advocate for the $R$-risk, Mahajan et al. (2023) favor the $T$- and $DR$-risk. Conversely, Curth & van der Schaar (2023) show that the effectiveness of different risk measures varies with various factors, such as the metalearner and data generating process, with no single criterion being universally optimal. Additionally, Doutreligne & Varoquaux (2023) stress the flexibility of the estimators used to construct the pseudo-labels, with Mahajan et al. (2023) recommending the use of AutoML. These complexities and design choices highlight the need for automated procedures.

> **Research gap**—Despite significant recent advances in ML for both CATE estimation and model validation, critical gaps remain in understanding when to use specific methods, how to effectively tune them, and how to address essential but overlooked aspects like preprocessing or ensembling.

## 3 PROBLEM FORMULATION

**Notation and assumptions.** We represent an instance by a tuple $(x, t, y)$, with covariates $X \in \mathcal{X} \subset \mathbb{R}^d$, a treatment $T \in \mathcal{T} = \{0, 1\}$, and an outcome $Y \in \mathcal{Y} \subset \mathbb{R}$. The potential outcome $Y$ associated with a treatment $t$ is denoted as $Y(t)$. We aim to estimate the conditional average treatment effect (CATE): $\tau = \mathbb{E}[Y(1) - Y(0)|X]$. Estimating the CATE from observational data requires standard assumptions (see Appendix A.2). More background on CATE estimation is provided in Appendix A.

---

[1]For example, consider the case where both potential outcomes are overestimated by the same amount. Even though $\mu$-risk would indicate a poor model quality, the resulting CATE estimates would still be accurate.

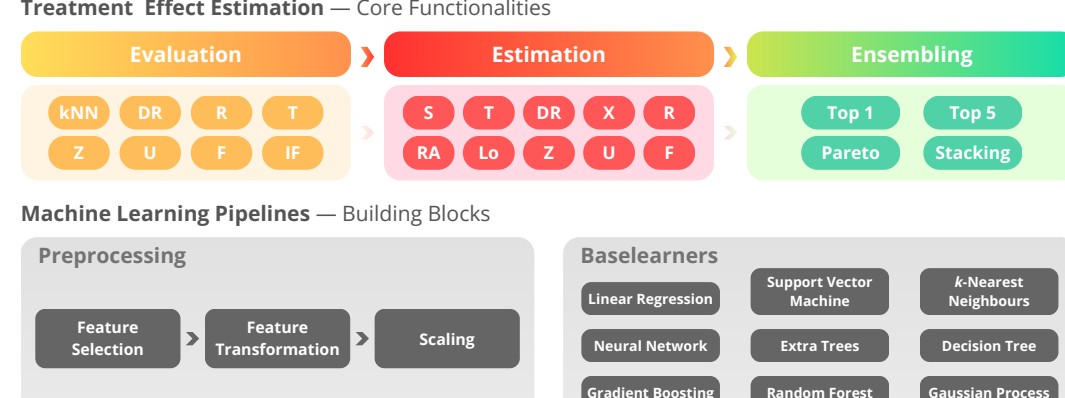

Figure 2: **AutoCATE overview.** We estimate treatment effects in three stages: (1) *Evaluation–* learning the appropriate risk measure(s), (2) *Estimation–*tuning a CATE estimation pipeline, and (3) *Ensembling–*selecting a final model or constructing an ensemble. We build ML pipelines for evaluation and estimation based on a collection of *preprocessing algorithms* and ML *baselearners*.

**Goals and challenges.** We aim to develop a *general procedure* for learning a pipeline for CATE estimation from an observational data set. Formally, this is a *counterfactual Combined Algorithm Selection and Hyperparameter optimization* (CASH) problem. It involves searching over ML pipelines $a_h$ with algorithms $a \in \mathcal{A}$ and hyperparameters $h \in \mathcal{H}_a$ to minimize the error on test data $\mathcal{D}_{\text{test}}$:

$$\arg\min_{a,h} \mathcal{L}(a_h|\mathcal{D}_{\text{test}}). \tag{1}$$

An algorithm $a$ can be an ML method tailored for CATE estimation or a metalearner combining one or more baselearners. Solving the counterfactual CASH problem involves several *unique challenges*. An algorithm's quality of fit on the train data $\mathcal{L}(a_h|\mathcal{D}_{\text{train}})$ is unobserved, as there is no ground truth CATE. Additionally, there is covariate shift between the observational training data and test data due to confounding. Both points present challenges for both *building* and *validating* an ML pipeline.

## 4 AutoCATE: End-To-End, Automated CATE Estimation

AutoCATE finds an optimal ML pipeline in three stages: *evaluation*, *estimation*, and *ensembling*.

(1) EVALUATION: In the first stage, we construct a proxy risk for $\mathcal{L}$ based on a risk measure (e.g., $R$-risk) and evaluation metric (e.g., MSE). To accurately estimate this risk on the validation data, we perform an automated search over preprocessors, ML algorithms, and their hyperparameters.

(2) ESTIMATION: The second stage automatically searches over combinations of preprocessors, metalearners, baselearners, and their hyperparameters to obtain ML pipelines for CATE estimation.

(3) ENSEMBLING: The final stage uses the proxy risk from the first stage to select and combine estimation pipelines from the second stage. The result can be a single ML pipeline or an ensemble.

A high-level overview of AutoCATE's functionalities and building blocks is shown in Figure 2.

### 4.1 Stage 1: Evaluation—Designing a Proxy Risk and Evaluation Protocol

The counterfactual CASH problem requires minimizing $\mathcal{L}(a_h|\mathcal{D}_{\text{test}})$, which involves two challenges: the lack of ground truth $\tau$ and the presence of covariate shift due to confounding. To tackle these, the evaluation stage measures risk by learning *pseudo-labels*–i.e., proxies for $\tau$–from validation data.

**Risk measures.** AutoCATE includes *different possible risk measures*, described in Appendix B.2. We include pseudo-labels used in metalearners ($DR$-, $R$-, $Z$-, $U$-, and $F$), plug-in risks ($T$ and $1NN$), and a risk approximation using influence functions ($IF$). We exclude the $\mu$- and $\mu_{\text{IPW}}$-risks as they do not apply to all metalearners, and the $S$-risk due to poor results in prior work (e.g., Mahajan et al., 2023). As constructing these risk measures requires accurately estimating nuisance parameters, we search over preprocessing and ML algorithms to find good-performing ML pipelines.

There is no ground truth, and different measures may be preferable depending on the (unknown) data generating process. To make our evaluation more robust, we allow for *combining different measures*. Similarly, since pseudo-outcomes are learned from data, there is no "true" version, enabling us to construct multiple version of a single risk (e.g., two $R$-risks). Using multiple risk measures results in a multi-objective search problem. To account for the varying scales of different risks, we normalize them by comparing each model's performance to an average treatment effect (ATE) baseline.

**Metrics and implementation.** Given a risk measure, different metrics can compare the pseudo-outcomes and CATE predictions to evaluate the quality of the ML pipeline. We include *general metrics* of predictive accuracy, like the mean squared error (MSE) or mean absolute percentage error (MAPE), and metrics related to a *downstream application*, such as the Area Under the Qini Curve (AUQC) when ranking effects (Vanderschueren et al., 2024). The $R$-risk requires a metric that accommodates weights. Finally, we allow for a stratified training-validation split or a stratified $k$-fold cross-validation procedure. Figure 8 shows more information on these evaluation frameworks.

### 4.2 STAGE 2: ESTIMATION—BUILDING A CATE ESTIMATION PIPELINE

Different *metalearners* can be used to estimate the CATE. Metalearners are general frameworks for using ML algorithms to estimate treatment effects. As such, they are versatile, accommodate various ML algorithms, and can be efficiently trained using existing ML packages. Common examples include the $S$-Learner (single model with the treatment as a feature), $Lo$-Learner (single model with treatment interaction terms), and $T$-Learner (separate models for each treatment group). Other metalearners use pseudo-outcomes that converge to the treatment effect, such as the $DR$-, $X$-, $R$-, $RA$-, $Z$-, $U$-, and $F$-Learners. Appendix B.1 provides more detailed information on each metalearner. Our package uses the `CausalML` implementations where available (Chen et al., 2020).

### 4.3 STAGE 3: ENSEMBLING—SELECTING AND ENSEMBLING ESTIMATION PIPELINES

The pipelines from the *estimation* stage are evaluated with risk measures from the *evaluation* stage. The final *ensembling* stage selects the best pipeline(s) for prediction. We describe different possible approaches here, with detailed descriptions provided in Appendix B.5. Almost no established methods exist for ensembling CATE estimators and, due to the lack of ground truth, most standard ensembling methods are not applicable. `AutoCATE` can select the best-performing pipeline or the top five for improved robustness and accuracy. We also include a novel stacking procedure that assigns weights (between zero and one) to each pipeline and optimizes these to minimize the squared error with respect to the pseudo-outcomes. The weights are regularized, with tuning on a holdout set. Finally, we also include the stacking procedure with softmax weights of Mahajan et al. (2023)–to the best of our knowledge, this is the only existing ensemble method tailored for CATE estimation.

With *multiple risk measures* in a multi-objective search, there may not be a single optimal pipeline, but rather a Pareto frontier. One strategy is to select all Pareto optimal points, though pipelines that perform very well on only a single measure may not work well generally. To select pipelines with good general performance, we can select the pipeline (or the top five) with the lowest average risk across objectives. Similarly, we can select based on each pipeline's Euclidean distance to the origin, or its average rank across objectives. Finally, we can apply the abovementioned stacking procedure for each risk measure separately and averaging the weights in a final stacked pipeline.

### 4.4 ML PIPELINE BUILDING BLOCKS: PREPROCESSING AND ML BASELEARNERS

We construct ML pipelines in both the *evaluation* and *estimation* stage. The building blocks for these include preprocessors and ML algorithms, all built on top of `scikit-learn` (Pedregosa et al., 2011). For *preprocessing*, we provide different feature selection and scaling algorithms. As *baselearners*, we include different ML algorithms with both classification and regression counterparts, ranging from linear regression to random forests. We provide more information in Appendix B.3.

The final search space includes a variety of preprocessors, metalearners, baselearners, and their hyperparameters. Efficient *optimization schemes* such as Bayesian optimization could be used, but we use random search throughout this work to focus on other design choices in `AutoCATE`. Nevertheless, we implement our search using `optuna` (Akiba et al., 2019), allowing easy integration of sophisticated optimizers like a Tree-structured Parzen Estimator (Bergstra et al., 2011).

### 4.5 LOW-CODE CATE ESTIMATION THROUGH AUTOCATE'S API

`AutoCATE` is implemented in Python[2], following `scikit-learn`'s design principles (Pedregosa et al., 2011). The low-code API enables automated CATE estimation with just four lines of code:

```
1  from src.AutoCATE import AutoCATE        # Import the AutoCATE class
2  autocate = AutoCATE()                    # Initialize the AutoCATE object
3  autocate.fit(X_train, t_train, yf_train) # Find the best pipeline(s)
4  cate_pred = autocate.predict(X_test)     # Predict the CATE for new data
```

Initialization arguments can be specified (e.g., the number of estimation trials; see Appendix B.6).

## 5 EMPIRICAL EVALUATION: COMPARING AUTOMATED STRATEGIES

This section empirically compares design choices for solving the counterfactual CASH problem for all three stages: *evaluation* (5.2), *estimation* (5.3), and *ensembling* (5.4). We identify best practices and benchmark the resulting configuration against common approaches for CATE estimation (5.5).

### 5.1 EXPERIMENTAL SETUP: DATA AND EVALUATION METRICS

Our experiments compare various automated, end-to-end strategies for learning a CATE estimation pipeline. Using `AutoCATE`, we evaluate design choices in each stage: evaluation, estimation, and ensembling. To obtain general insights, we leverage a collection of standard benchmarks for CATE estimation: IHDP (Hill, 2011), ACIC Dorie et al. (2019), News (Johansson et al., 2016), and Twins (Louizos et al., 2017); see Appendix C for details. These semi-synthetic benchmarks include 247 distinct data sets that vary in outcome (regression and classification), dimensionality, size, and application area, allowing for a comprehensive analysis `AutoCATE`. Unless noted otherwise, results are reported in precision in estimating heterogeneous treatment effects (PEHE): $\sqrt{\text{PEHE}} = \sqrt{(\tau - \hat{\tau})^2}$.

For each experimental result, the caption clarifies the `AutoCATE` configuration used. For the evaluation and estimation stages, we describe the search strategy for automatically optimizing the ML pipelines, including base- and metalearners involved and the number of optimization trials per stage. Unless stated otherwise, `AutoCATE` select the best ML pipeline based on best average performance.

### 5.2 ANALYZING AUTOCATE—STAGE 1: EVALUATION PROTOCOL

We analyze the evaluation protocol by comparing risk measures, metrics, and evaluation procedures.

#### 5.2.1 HOW TO MEASURE RISK REGARDING CATE PREDICTIONS?

**What risk measure works best?** We compare predictive error resulting from model selection with different risk measures in Table 1a. Three options consistently show low error: the *DR*-, *kNN*-, and *T*-risk. These results largely correspond with existing work. Curth & van der Schaar (2023); Mahajan et al. (2023) similarly found the *DR*-risk to work well, though the *kNN*-risk works comparatively better in our experiments. Although Curth & van der Schaar (2023) reported worse results for the *T*-risk, both our findings and those in Mahajan et al. (2023) show that it *can* give good results with proper tuning of the underlying models. We further analyze the impact of tuning in Figure 3: increased tuning for the evaluation models generally results in better downstream performance. To test whether congeniality bias affects our results (Curth & van der Schaar, 2023), we repeat this experiment for different metalearners in Table 7. Again, the *T*-, *DR*-, and *kNN*-risk perform best.

**Is it beneficial to use multiple risk measures?** We explore the impact of combining different risk measures in a multi-objective search, hypothesizing that this could lead to more robust pipeline selection as each measure is a different proxy to the same ground truth. Table 1b shows both results for risk measure combinations, and for multiple versions of a single measure based on different estimates. We observe that combining different types or different versions of risk measures can indeed improve performance, though no strategy substantially improves upon the best single measure.

---

[2]Our package and experimental code are available at https://anonymous.4open.science/r/AutoCATE-E103.

| | DR | F | IF | kNN | R | T | U | Z |
|---|---|---|---|---|---|---|---|---|
| *IHDP* | **2.12**$_{\pm.34}$ | 3.33$_{\pm.55}$ | 3.13$_{\pm.45}$ | 2.22$_{\pm.36}$ | 3.37$_{\pm.71}$ | 2.15$_{\pm.35}$ | 3.58$_{\pm.72}$ | 5.40$_{\pm.86}$ |
| *ACIC* | 1.56$_{\pm.09}$ | 1.74$_{\pm.10}$ | 2.52$_{\pm.16}$ | 1.74$_{\pm.10}$ | 1.63$_{\pm.10}$ | **1.52**$_{\pm.09}$ | 1.72$_{\pm.09}$ | 2.40$_{\pm.15}$ |
| *Twins* | .333$_{\pm.00}$ | .340$_{\pm.00}$ | .340$_{\pm.01}$ | .323$_{\pm.00}$ | .335$_{\pm.00}$ | **.323**$_{\pm.00}$ | .359$_{\pm.01}$ | .350$_{\pm.01}$ |
| *News* | **2.42**$_{\pm.07}$ | 2.48$_{\pm.07}$ | 2.73$_{\pm.09}$ | 2.43$_{\pm.07}$ | 2.51$_{\pm.08}$ | 2.42$_{\pm.07}$ | 2.60$_{\pm.09}$ | 3.02$_{\pm.11}$ |

(a) Comparing downstream performance for different risk measures

| | **Combining risks** | | | $T$-**risk—Multiple versions** | | | | **Best** |
|---|---|---|---|---|---|---|---|---|
| | *All* | *DR,T* | *DR,T,kNN* | *Top 1* | *Top 2* | *Top 3* | *Top 5* | **single** |
| *IHDP* | 2.48$_{\pm.36}$ | 2.19$_{\pm.35}$ | 2.13$_{\pm.35}$ | 2.15$_{\pm.35}$ | 2.15$_{\pm.35}$ | 2.17$_{\pm.35}$ | **2.11**$_{\pm.36}$ | 2.12$_{\pm.34}$ |
| *ACIC* | 1.94$_{\pm.13}$ | 1.58$_{\pm.09}$ | 1.60$_{\pm.09}$ | **1.52**$_{\pm.09}$ | 1.54$_{\pm.08}$ | 1.55$_{\pm.09}$ | **1.52**$_{\pm.08}$ | 1.52$_{\pm.09}$ |
| *Twins* | .331$_{\pm.01}$ | **.323**$_{\pm.00}$ | .324$_{\pm.00}$ | **.323**$_{\pm.00}$ | 0.323$_{\pm.00}$ | **.323**$_{\pm.00}$ | .324$_{\pm.00}$ | **.323**$_{\pm.00}$ |
| *News* | 2.52$_{\pm.07}$ | 2.41$_{\pm.06}$ | **2.41**$_{\pm.07}$ | 2.42$_{\pm.07}$ | 2.41$_{\pm.07}$ | 2.43$_{\pm.07}$ | 2.43$_{\pm.07}$ | 2.42$_{\pm.07}$ |

(b) Comparing downstream performance for different combinations of risk measures

Table 1: **Performance for validation based on different risk measures.** Results in $\sqrt{\text{PEHE}}_{\pm\text{SE}}$ (lower is better). **Bold** highlights the best results, with underlined values falling within 1 standard error. Results for 50 evaluation trials and 50 estimation trials with a $T$-Learner and gradient boosting.

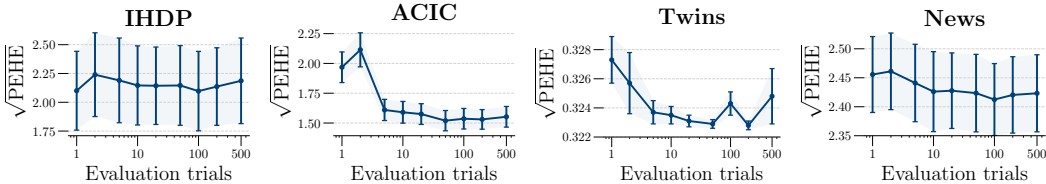

Figure 3: **How many iterations should we tune evaluation models?** We compare downstream results, based on different number of trials used to tune the models underlying the evaluation metrics. Results for a $T$-risk and 50 estimation trials with a $T$-Learner and gradient boosting.

### 5.2.2 WHAT EVALUATION PROCEDURE TO USE?

**How to set the holdout ratio?** Risk measures require estimates learned from validation data, creating a *trade-off* between using data for evaluation or estimation. Figure 4 presents results for different holdout ratios, illustrating this trade-off and showing that a holdout ratio of 30-50% generally works well. We use 30% for holdout in the rest of this work. Although more folds in cross-validation often improve model performance in supervised settings, we do not observe this effect for `AutoCATE` (see Table 6), likely due to the complex interplay between the number of folds and the holdout ratio.

**What evaluation metric to use?** All previous experiments used the mean squared error (MSE) to compare the predicted CATE and pseudo-outcome(s), corresponding with the goal of minimizing PEHE. However, depending on the downstream application, *alternative metrics* might be more important. Using these in `AutoCATE` is straightforward. Table 2 shows results for two such metrics: the mean absolute percentage error (MAPE) and area under the Qini curve (AUQC). As hypothesized, selecting models based on a particular metric generally improves performance for that metric.

### 5.3 ANALYZING AutoCATE—STAGE 2: ESTIMATION PROTOCOL

Given an *evaluation* protocol, we can compare strategies for the *estimation* stage. This section examines how including different metalearners and baselearners affects `AutoCATE`'s performance.

**Metalearners.** Figure 5 compares different versions of `AutoCATE` with either all meta- and base-learners (see Figure 2 for an overview), or only the best per category. The complete "AllMeta-AllBase" sometimes performs poorly. While performance generally improves with more trials, poor results persists even after 100 trials on the News data. Further inspection reveals that bad iterations are due to instability of the $R$- and $U$-Learners: these are chosen due to good initial performance on the validation set, but can perform exceptionally poor on the test data after retraining on all data. Other metalearners ($F$ and $Z$) are *almost never* chosen. Therefore, "BestMeta" excludes these

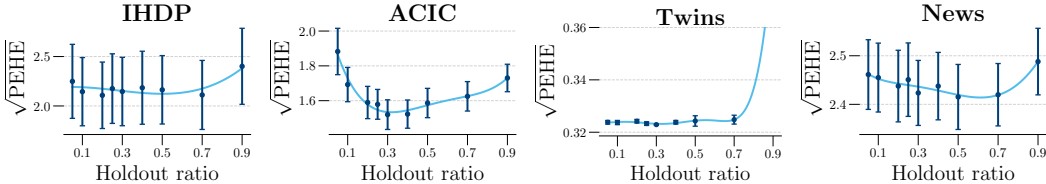

Figure 4: **How much data to use for evaluation?** We show results for different holdout ratios and fit a polynomial function for each data set to gain insight into the optimal ratio. Results for 50 evaluation trials with a $T$-risk and 50 estimation trials with a $T$-Learner and gradient boosting.

|  | MSE | MAPE | AUQC |  |  | MSE | MAPE | AUQC |
|---|---|---|---|---|---|---|---|---|
| $\sqrt{\text{PEHE}}$ | $\mathbf{2.15}_{\pm 0.35}$ | $\underline{2.28}_{\pm .36}$ | $\underline{2.26}_{\pm .41}$ |  | $\sqrt{\text{PEHE}}$ | $\underline{1.52}_{\pm .09}$ | $1.67_{\pm .09}$ | $\mathbf{1.50}_{\pm .08}$ |
| MAPE | $1.76_{\pm 1.30}$ | $1.40_{\pm .94}$ | $\mathbf{0.50}_{\pm .15}$ |  | MAPE | $\underline{1.10}_{\pm .21}$ | $\mathbf{1.03}_{\pm .14}$ | $\underline{1.11}_{\pm .24}$ |
| AUQC | $0.92_{\pm 0.01}$ | $0.88_{\pm .02}$ | $\mathbf{0.96}_{\pm .01}$ |  | AUQC | $\underline{0.91}_{\pm .01}$ | $0.90_{\pm .01}$ | $\mathbf{0.91}_{\pm .01}$ |

(a) IHDP  (b) ACIC

|  | MSE | MAPE | AUQC |  |  | MSE | MAPE | AUQC |
|---|---|---|---|---|---|---|---|---|
| $\sqrt{\text{PEHE}}$ | $\mathbf{.323}_{\pm .00}$ | $.323_{\pm .00}$ | $.344_{\pm .00}$ |  | $\sqrt{\text{PEHE}}$ | $\mathbf{2.42}_{\pm .07}$ | $2.52_{\pm .07}$ | $\underline{2.46}_{\pm .07}$ |
| MAPE | — | — | — |  | MAPE | $\mathbf{5.75}_{\pm .74}$ | $\underline{5.83}_{\pm .69}$ | $\underline{5.86}_{\pm .85}$ |
| AUQC | $0.00_{\pm .00}$ | $0.00_{\pm .01}$ | $\mathbf{0.03}_{\pm .01}$ |  | AUQC | $\mathbf{0.66}_{\pm .01}$ | $0.64_{\pm .01}$ | $\underline{0.65}_{\pm .01}$ |

(c) Twins  (d) News

Table 2: **Comparing different evaluation metrics.** We compare model selection with different evaluation metrics. For the Twins data set, MAPE cannot be calculated, as the true CATE can be zero. **Bold** highlights the best results, with underlined values falling within 1 standard error. Colored cells show the hypothesis that matching metrics will yield the best performance. Results for 50 evaluation trials with a $T$-risk and 50 estimation trials with a $T$-Learner and gradient boosting.

metalearners ($R$, $F$, $Z$, and $U$), resulting in improved stability and performance. Appendix D.2 compares metalearners' precision and time efficiency, and shows how often metalearners are chosen.

**Baselearners.** The "BestBase" versions in Figure 5 only use base learners that typically perform well with tabular data: random forests, extremely randomized trees, gradient boosting, and multilayer perceptrons. This constraint is applied to both evaluation and estimation pipelines. While selecting these baselearners improves performance, it is less significant than filtering metalearners.

5.4 ANALYZING AUTOCATE—STAGE 3: ENSEMBLING PROTOCOL

The *ensemble* stage compares pipelines built in the estimation stage using the objective(s) learned in the evaluation stage. Selected pipelines are re-trained on the entire data and saved for inference.

**Single objective.** With a single objective, we can select the best pipeline (Top 1), the best five (Top 5), or use stacking to build a final estimator that combines all pipelines. Table 3a compares these strategies, showing that *combining pipelines improves performance* for all data sets except Twins. Appendix D.3 illustrates how an ensemble's predictions can help assess an estimate's uncertainty.

**Multiple objectives.** Model selection is more complex with multiple objectives. We can select the best pipelines based on the average normalized score, Euclidean distance to the origin, or average rank, to then select the top one or top five pipelines. Alternatively, we can create stacking estimators for each objective and average the weights ("Stacking"), or select all Pareto optimal models ("Pareto"). Table 3b compares these strategies. Single pipelines typically underperform compared to ensembles built from the top five pipelines, all Pareto optimal pipelines, or stacking. Selecting based on average performance yields the best performance. No single strategy is consistently optimal.

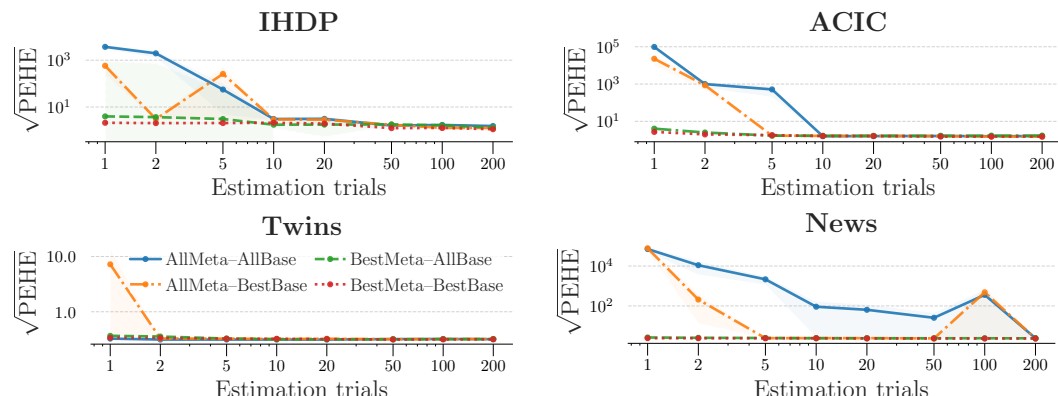

Figure 5: **What meta- and baselearners to include?** We compare different search spaces for `AutoCATE`, either including all metalearners (AllMeta) or only the best (BestMeta), as well as all baselearners (AllBase) or only the best (BestBase). Results for 50 evaluation trials with a $T$-risk.

|  | **Best model(s)** | | **Stacking** | |
|---|---|---|---|---|
|  | *Top 1* | *Top 5* | *COP* | *Softmax* |
| *IHDP* | $\underline{2.15}_{\pm.35}$ | $\mathbf{1.90}_{\pm.34}$ | $\underline{1.96}_{\pm.34}$ | $2.83_{\pm.51}$ |
| *ACIC* | $1.52_{\pm.09}$ | $\underline{1.34}_{\pm.08}$ | $\underline{1.42}_{\pm.09}$ | $\mathbf{1.33}_{\pm.09}$ |
| *Twins* | $\mathbf{.323}_{\pm.00}$ | $.325_{\pm.00}$ | $.344_{\pm.00}$ | $.331_{\pm.00}$ |
| *News* | $2.42_{\pm.07}$ | $\underline{2.33}_{\pm.06}$ | $\underline{2.33}_{\pm.06}$ | $\mathbf{2.32}_{\pm.06}$ |

(a) Comparing ensemble strategies for a single $T$-risk

|  | **Average** | | **Distance** | | **Ranking** | | **Stacking** | | |
|---|---|---|---|---|---|---|---|---|---|
|  | *Top 1* | *Top 5* | *Top 1* | *Top 5* | *Top 1* | *Top 5* | *COP* | *Softmax* | **Pareto** |
| *IHDP* | $2.19_{\pm.35}$ | $\mathbf{1.84}_{\pm.31}$ | $2.27_{\pm.37}$ | $2.99_{\pm.54}$ | $3.58_{\pm.66}$ | $2.99_{\pm.54}$ | $\underline{1.94}_{\pm.32}$ | $2.83_{\pm.51}$ | $2.19_{\pm.36}$ |
| *ACIC* | $1.58_{\pm.09}$ | $\underline{1.35}_{\pm.08}$ | $1.55_{\pm.08}$ | $\underline{1.41}_{\pm.08}$ | $1.69_{\pm.08}$ | $\underline{1.41}_{\pm.08}$ | $1.43_{\pm.09}$ | $\mathbf{1.33}_{\pm.09}$ | $1.50_{\pm.08}$ |
| *Twins* | $\mathbf{.323}_{\pm.00}$ | $.325_{\pm.00}$ | $\mathbf{.323}_{\pm.00}$ | $.341_{\pm.00}$ | $.367_{\pm.01}$ | $.341_{\pm.00}$ | $.349_{\pm.00}$ | $.331_{\pm.00}$ | $.326_{\pm.00}$ |
| *News* | $2.41_{\pm.06}$ | $\underline{2.32}_{\pm.06}$ | $2.42_{\pm.07}$ | $\underline{2.38}_{\pm.07}$ | $2.58_{\pm.08}$ | $\underline{2.38}_{\pm.07}$ | $\underline{2.34}_{\pm.06}$ | $\mathbf{2.32}_{\pm.06}$ | $2.39_{\pm.07}$ |

(b) Comparing ensemble strategies when combining $DR$- and $T$-risks

Table 3: **Ensemble strategies.** We compare ensembling strategies for a single or multiple objectives in terms of $\sqrt{\text{PEHE}}$. **Bold** highlights the best results, underlined values lie within 1 standard error. Results for 50 evaluation trials and 50 estimation trials with a $T$-Learner and gradient boosting.

## 5.5 BENCHMARKING AuToCATE AGAINST COMMON ALTERNATIVES

This section compares the optimized configuration of `AutoCATE` with some common alternative approaches for tuning CATE estimation pipelines. These benchmarks select the best model using the error in predicting observed outcomes ($\mu$-risk). We include both $S$- and $T$-Learners. For $T$-Learners, we tune models separately for the control and treatment groups. First, we compare a $T$-Learner with gradient boosting tuned based on the $\mu$-risk against `AutoCATE` using only a $T$-Learner and gradient boosting optimized for $T$-risk. While these strategies are similar, `AutoCATE` evaluates the entire pipeline jointly and (potentially) adds preprocessing. Conversely, the traditional $T$-Learner's search is more efficient as it tunes models separately per group. Figure 6 compares the two approaches: the $\mu$-risk strategy performs worse for Twins, but better for ACIC. Finally, Figure 7 compares `AutoCATE` with $S$- and $T$-Learners using random forests and gradient boosting. These approaches are conceptually simple, but represent common and strong baselines. We observe that, for each data set, `AutoCATE` can obtain at least competitive performance to the best approach. These strong results are due to two factors. First, `AutoCATE` offers greater flexibility through a larger search space, including more meta- and baselearners and preprocessing (Table 10 analyzes the added value of preprocessing). Second, model selection is better aligned with the goal of CATE

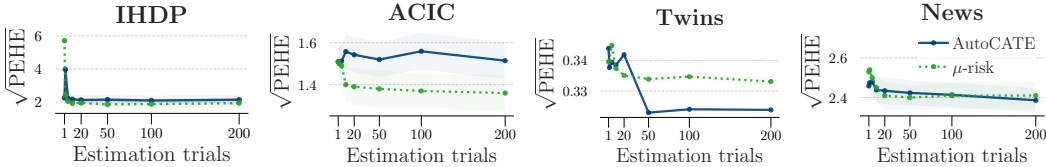

Figure 6: **Comparing AutoCATE with tuning based on $\mu$-risk.** We compare tuning a $T$-Learner with gradient boosting using either AutoCATE (based on a $T$-risk) or tuning based on the MSE on the observed outcome. AutoCATE uses a $T$-risk with 50 evaluation trials and top 1 model selection.

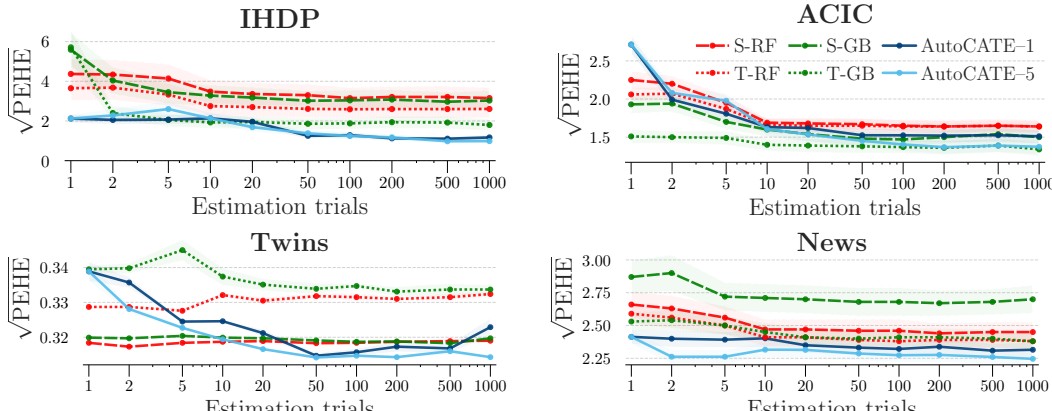

Figure 7: **Benchmarking AutoCATE.** We compare AutoCATE with common benchmarks using $S$- and $T$-Learners with random forests and gradient boosting. AutoCATE uses a $T$-risk with 50 evaluation trials and BestMeta-BestBase search spaces, with either Top 1 or Top 5 model selection.

estimation, using the $T$-risk, and can include an ensemble of pipelines for improved performance. Appendix D.4 shows similar results for ranking treatment effects with data from uplift modeling.

## 6 CONCLUSION

Despite the availability of ML methods for CATE estimation, their *adoption remains limited*, due to the complexity of implementing, tuning, and validating them. We framed the problem of finding an ML pipeline for CATE estimation as a *counterfactual CASH problem* and proposed AutoCATE: the first *end-to-end, automated solution* tailored for treatment effect estimation. Based on this solution, we analyzed design choices for evaluation, estimation, and ensembling, and identified best practices. The resulting approach was validated, outperforming widely used strategies for CATE estimation.

To maximize AutoCATE's practical impact, several *limitations* need to be addressed. Although AutoCATE relies on standard *assumptions* for causal inference, it is crucial to assess its robustness against violations of these assumptions and potentially protocols for such scenarios. Additionally, most of the data used in this work is *semi-synthetic* (IHDP, ACIC, and News), which may not fully capture the complexities of real-world data. Although validating CATE estimates remains inherently challenging, approaches from related fields could offer inspiration (see e.g. Devriendt et al., 2020).

AutoCATE enables a *comprehensive analysis* of existing methods (see Figure 1 and Appendix D.5), facilitating a better understanding of CATE estimation and guiding the development of new approaches. We envision opportunities for *future research* in all stages. For *evaluation*, advanced multi-objective strategies could improve performance and robustness. Novel methods for *estimation* could be automatically discovered using Neural Architecture Search. Generally, efficiency can be improved with better search algorithms or strategies (e.g., by re-using nuisance models across metalearners). Related to this, the optimal time allocation between the stages remains an open question, where meta-learning could help by incorporating data set characteristics (Feurer et al., 2015). Finally, more advanced *ensembling* could be developed (e.g., combining different metalearners).

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

The appendix starts with a more detailed introduction and background to CATE estimation in Appendix A. The next sections provide more details on `AutoCATE` (Appendix B), describe the data sets used in this work (Appendix C), and present additional empirical results (Appendix D). Finally, we compare `AutoCATE` with other packages for CATE estimation in Appendix E.

## A  BACKGROUND ON CATE ESTIMATION

This section provides a more detailed introduction and background on treatment effect estimation. In accordance to the main body, we denote an instance by a tuple $(x, t, y)$, with covariates $X \in \mathcal{X} \subset \mathbb{R}^d$, a treatment $T \in \mathcal{T} = \{0, 1\}$, and an outcome $Y \in \mathcal{Y} \subset \mathbb{R}$. Following the potential outcomes framework (Rubin, 1974; 2005), we describe an instance's potential outcome $Y$ for a given treatment $T = t$ as $Y(t)$. The Conditional Average Treatment Effect (CATE) is then defined as the expected difference in outcomes between treating and not treating:

$$\mathbb{E}\Big[Y(1) - Y(0)|X\Big]. \tag{2}$$

Knowing this effect is crucial in a variety of domains, such as education (Olaya et al., 2020), healthcare (Feuerriegel et al., 2024), and maintenance (Vanderschueren et al., 2023). Estimating the CATE from observational data involves significant *challenges* (Appendix A.1), requires standard *assumptions* (Appendix A.2), and tailored ML methods (Appendix A.3). We explain these in the following.

### A.1  CHALLENGES: THE FUNDAMENTAL PROBLEM AND CONFOUNDING

The fundamental problem of causal inference (Holland, 1988) is that, for each instance, we only observe either $Y(0)$ or $Y(1)$, depending on what treatment was administered. We refer to the observed outcome as the factual outcome and the unobserved outcome as the counterfactual outcome. Because one outcome is always unobserved, we never know the true CATE $\tau$, which means that there is *no ground truth* CATE available for training or validation.

In observational data, the outcome that was observed is typically not random: some instances were more likely to be treated, while other instances were more likely not to receive treatment. For example, in healthcare, patients may be more likely to receive a new treatment if they have access to better healthcare, have no pre-existing conditions, and are younger. The covariates that influence both the outcome and treatment assignment are called *confounders*, with the resulting non-random treatment assignment sometimes referred to as confounding.

Confounding presents an additional challenge for CATE estimation and validation as it results in *covariate shift*. Some instance-treatment pairs (the counterfactuals) will be absent in the observational training data compared to the hypothetical test data that contains all instance-treatment pairs (both factuals and counterfactuals). Because of this, an ML model may focus too much on the observed data points at the cost of worse predictions for the counterfactuals and, as such, the test data overall.

### A.2  ASSUMPTIONS FOR IDENTIFIABILITY

Identifying the causal effect from observational data requires making standard assumptions: consistency, overlap, and unconfoundedness. This section explains these assumptions in more detail.

**Assumption 1 (Consistency)** *The observed outcome given a treatment is the potential outcome under that treatment: $Y|X, t = Y(t)|X$.*

**Assumption 2 (Overlap)** *For each instance, there is a non-zero probability of receiving each treatment given their covariates: $\forall\, x \in \mathcal{X}$ and $t \in \mathcal{T} : P(T = t|X = x) > 0$. This condition ensures that there is sufficient variability in the treatment assignment.*

**Assumption 3 (Unconfoundedness)** *Given an instance's covariates, its potential outcomes are independent of the treatment assignment: $Y(0), Y(1) \perp\!\!\!\perp T \mid X$. This condition implies that all factors influencing both the treatment assignment and outcome are included in $X$. In other words, there are no unobserved confounders.*

There has recently been much interest in CATE estimation under violation of these assumptions. For example, by quantifying the uncertainty or sensitivity of an estimate to a possible violation (Franks et al., 2020; Jesson et al., 2020; 2021), characterizing overlap violations (Oberst et al., 2020), or developing metalearners that can deal with unobserved confounders (Oprescu et al., 2023). We believe that extending `AutoCATE` to deal with these settings and to incorporate these methods will improve its potential for real-world applicability even further. As such, we consider it an important direction for future versions.

### A.3 CATE ESTIMATION: META- AND BASELEARNERS

We briefly describe the approach of estimating the CATE with a metalearner here. A straightforward way of estimating the CATE is using a single ML model, where the treatment variable is considered an ordinary input variable. This metalearner is called the $S$-Learner and can be implemented with a wide variety of baselearners (i.e., ML algorithms that predict an outcome based on data, such as a decision tree or neural network). An alternative metalearner, the $T$-learner, fits two models–one model for each treatment group. Both models can use the same baselearner or a different one. More information on the metalearners in `AutoCATE` is provided in Appendix B.1. For more extensive overviews, we refer to Devriendt et al. (2018), Zhang et al. (2021), and Feuerriegel et al. (2024).

## B  AUTOCATE: ADDITIONAL INFORMATION

This section presents information on metalearners (Appendix B.1), risk measures for evaluation (Appendix B.2), and `AutoCATE`'s search spaces for preprocessors and baselearners (Appendix B.3).

### B.1  METALEARNERS

We describe the metalearners implemented in `AutoCATE` in more detail below. We first define the estimates that make up the building blocks of these models: the estimated propensity score $\hat{e}(x) = \mathbb{E}(t|x)$, the treatment-group specific outcome $\hat{y}_0(x) = \mathbb{E}(y|x, t = 0)$ and $\hat{y}_1(x) = \mathbb{E}(y|x, t = 1)$, and the treatment-unaware outcome $\hat{\mu}(x) = \mathbb{E}(y|x)$. In the following, the function $f$ describes a model that is learned with a base learner such as a neural network or gradient boosting.

**$S$-Learner.**  The $S$-Learner, or *single* learner, simply uses the treatment as a variable: $f_S(x, t) = \mathbb{E}(y|x, t)$. The CATE $\tau$ is then estimated as $\hat{\tau} = \hat{y}_1 - \hat{y}_0 = f_S(x, t = 1) - f_S(x, t = 0)$.

**$Lo$-Learner (Lo, 2002).**  The $Lo$-Learner is similar to an $S$-Learner, in the sense that it uses the treatment as a variable, but it adds interaction terms between the covariates $x$ and treatment $t$: $f_{Lo}(x, t) = \mathbb{E}(y|x, t, x \cdot t)$. The CATE $\tau$ is then estimated as $\hat{\tau} = \hat{y}_1 - \hat{y}_0 = f_{Lo}(x, t = 1) - f_{Lo}(x, t = 0)$.

**$T$-Learner.**  The $T$-Learner constructs *two* models–one per treatment group: $f_T^0(x) = \mathbb{E}(y|x, t = 0)$ and $f_T^1(x) = \mathbb{E}(y|x, t = 1)$, and predicts the CATE as $\hat{\tau} = \hat{y}_1 - \hat{y}_0 = f_T^1(x) - f_T^0(x)$.

**$X$-Learner (Künzel et al., 2019).**  The X-Learner first learns two treatment-specific outcome models: $\hat{y}_0(x)$ and $\hat{y}_1(x)$. It then uses these to impute the counterfactual outcome for each instance and, as such, obtain a pseudo-outcome $\tilde{\tau}_X$ for the treatment effect: $\tilde{\tau}_X^0 = \hat{y}_1(x) - y$ if $t = 0$, and $\tilde{\tau}_X^1 = y - \hat{y}_0(x)$ else. For each treatment group, a model is then learned on these pseudo-outcome: $f_X^0(x) = \tilde{\tau}_X^0$ and $f_X^1(x) = \tilde{\tau}_X^1$. The final effect model then estimates $f_X(x) = g(x)f_X^0 + (1 - g(x))f_x^1$ and predicts the treatment effect as $\hat{\tau} = f_X(x)$. $g(x) \in [0, 1]$ is a weighting function, typically the estimated propensity score $g(x) = \hat{e}(x)$.

**$RA$-Learner (Curth & van der Schaar, 2021).**  The $RA$-Learner or *regression-adjusted* learner is similar to an $X$-Learner, but directly learns the final model on the pseudo-outcomes: $f_{RA}(x) = \mathbb{E}(\tilde{\tau}_X|x)$, predicting the treatment effect as $\hat{\tau} = f_{RA}(x)$.

**$Z$-Learner.**  The transformed outcome approach (Jaskowski & Jaroszewicz, 2012; Powers et al., 2018) or inverse propensity weighted estimator (Curth & van der Schaar, 2021) uses a pseudo-

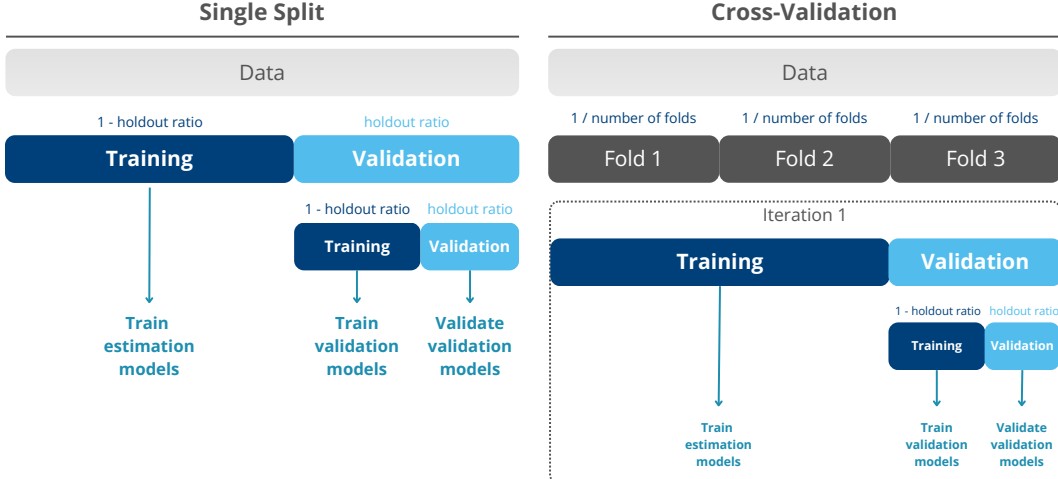

Figure 8: **Evaluation framework.** We show two possible frameworks for validating pipelines based on a single split or a cross-validation procedure. For each, the data is split in three groups to (1) train the estimation pipelines, (2) train the validation pipelines, and (3) validate the validation pipelines.

outcome based on the Horvitz-Thompson transformation (Horvitz & Thompson, 1952): $\tilde{\tau}_Z = \left(\frac{t}{\hat{e}(x)} - \frac{1-t}{1-\hat{e}(x)}\right) y$. The $Z$-Learner then estimates $f_Z(x) = \mathbb{E}(\tilde{\tau}_Z|x)$ and predicts the treatment effect as $\hat{\tau} = f_Z(x)$.

$U$**-Learner.** The $U$-Learner is based on a pseudo-outcome $\tilde{\tau}_U = \frac{y-\hat{\mu}(x)}{t-\hat{e}(x)}$. The final model fits $f_U(x) = \mathbb{E}(\tilde{\tau}_U|x)$ and predicts the treatment effect as $\hat{\tau} = f_U(x)$.

$F$**-Learner (Athey & Imbens, 2015).** The $F$-Learner uses the pseudo-outcome $\tilde{\tau}_F = \frac{t-\hat{e}(x)}{\hat{e}(x)(1-\hat{e}(x))} y$. The final model fits $f_F(x) = \mathbb{E}(\tilde{\tau}_F|x)$ and predicts the treatment effect as $\hat{\tau} = f_F(x)$.

$DR$**-Learner (Kennedy, 2023).** The $DR$-Learner is a robust version of the $Z$-Learner, based on the pseudo-outcome $\tilde{\tau}_Z = \left(\frac{t}{\hat{e}(x)} - \frac{1-t}{1-\hat{e}(x)}\right) y + \left(1 - \frac{t}{\hat{e}(x)}\right) \hat{y}_1(x) - \left(1 - \frac{1-t}{1-\hat{e}(x)}\right) \hat{y}_0(x)$. The final model is $f_{DR}(x) = \mathbb{E}(\tilde{\tau}_{DR}|x)$ and predicts the treatment effect as $\hat{\tau} = f_{DR}(x)$.

$R$**-Learner (Nie & Wager, 2021).** The $R$-Learner, based on Robinson's decomposition (Robinson, 1988), fits a model $f_R(x)$ using a weighted loss function with pseudo-outcomes $\tilde{\tau}_R = \frac{y-\hat{\mu}(x)}{t-\hat{e}(x)}$ and weights $w = (t - \hat{e}(x))^2$. The treatment effect can then directly be predicted as $\hat{\tau} = f_R(x)$.

### B.2 EVALUATION AND RISK MEASURES

The evaluation framework and data splitting underlying AutoCATE is shown in Figure 8. Below, we describe the different types of risk measures included in our framework.

**Metalearner pseudo-outcomes.** An instance's true CATE $\tau$ is unknown, but we can use the pseudo-outcomes $\tilde{\tau}$ used by the $T$-, $Z$-, $U$-, $F$-, $DR$-, and $R$-Learners (see above) as ground truth.

**Influence Function (IF) (Alaa & van der Schaar, 2019).** The influence function criterion gives an estimate of an ML pipeline's estimation error. It is based on a pseudo-outcome of the treatment effect $\tilde{\tau}$, estimated with a $T$-Learner. This pseudo-outcome is then debiased using the influence function. The final criterion is:

$$(1 - B)\,\tilde{\tau}^2 + By(\tilde{\tau} - \hat{\tau}) - D(\tilde{\tau} - \hat{\tau})^2 + \tilde{\tau}^2$$

with $D = t - \hat{e}(x)$, $C = \hat{e}(x)(1 - \hat{e}(x))$, and $B = 2tDC^{-1}$.

| Hyperparameter | Range |
|---|---|
| *VarianceThreshold* | |
| threshold | $[0, 0.04]$ |
| *SelectPercentile* | |
| k | $[5, \text{n\_dim}]$ |
| score_func | mutual_info_{regression, classif} |

(a) Feature Selection

| Hyperparameter | Range |
|---|---|
| *StandardScaler* | |
| — | |
| *RobustScaler* | |
| — | |

(b) Feature Scaling

Table 4: **Preprocessor search spaces.** We describe the search spaces for the different preprocessors. If a hyperparameter is not mentioned, we use its default. All preprocessors are implemented with scikit-learn (Pedregosa et al., 2011); we refer to their documentation for more information.

$k$**-Nearest Neighbor ($k$NN) (Rolling & Yang, 2014).** The nearest neighbor matching measure finds the nearest neighbor in the opposite group, defined using the Euclidean distance, and uses its outcome as the counterfactual outcome. As such, it is essentially a $T$-Learner pseudo-outcome where the baselearner is restricted to a nearest neighbor model. We extend upon this by allowing alternative versions to be constructed by increasing $k$.

### B.3 PREPROCESSOR AND BASELEARNER SEARCH SPACES

**Preprocessors.** ML pipelines include three (optional) steps to preprocess the data before being fed to a model: feature selection, transformation, and scaling. For feature selection, include VarianceThreshold, SelectPercentile, or no selection. For feature scaling, we include StandardScaler, RobustScaler, or no scaling. Finally, we include feature transformation algorithms in our software package (SplineTransformer, PolynomialFeatures, KBinsDiscretizer), but do not include them in the experiments as they significantly slowed down training times. Other steps for feature selection and scaling from *scikit-learn* are similarly supported, but not included in the experiments, which is why we do not discuss them here. Table 4 provides detailed information on the search spaces.

**Baselearners.** We present the search spaces for all baselearners' hyperparameters in Table 5. These are based largely upon existing AutoML packages (e.g., FLAML (Wang et al., 2021)) and some (limited) experimentation, so these may be improved in future versions.

AutoCATE's resulting search space of ML pipelines for CATE estimation is vast, with 2,187 possible pipelines even *without considering hyperparameters*:

$$3 \text{ feature selection} \times 3 \text{ scaling} \times 27 \text{ metalearner-baselearner configurations} \times 9 \text{ baselearners} \quad (3)$$

with $27 = 1 \ (S) + 2 \ (T) + 4 \ (DR) + 5 \ (X) + 4 \ (R) + 3 \ (RA) + 1 \ (Lo) + 2 \ (Z) + 3 \ (U) + 2 \ (F)$, i.e., the sum of all baselearners required per metalearner.

### B.4 EXAMPLE ML PIPELINE

We give an example of a pipeline built by AutoCATE, excluding baselearner hyperparameters. *Evaluation* using a $T$-Risk evaluation, with control outcomes estimated with gradient boosting and treatment outcomes estimated using a neural network. *Estimation* by first selecting a top percentile of features based on the F-value between the label and feature, followed by a $DR$-Learner where propensity scores are estimated with a support vector machine, control outcomes with gradient boosting, treatment outcomes with a linear regression, and the final effect with a random forest. This example illustrates the complexity of an ML pipeline for CATE estimation–in this case, there are six different ML models with several hyperparameters each. If an *ensemble* is used for estimation, this complexity increases even more.

### B.5 ENSEMBLING AND MULTI-OBJECTIVE MODEL SELECTION

This section describes the different approaches for ensembling and multi-objective model selection included in our framework. With multiple objectives, no globally optimal ML pipeline may exist. We explore various strategies for ranking and selecting models in this context. We denote a pipeline

| Hyperparameter | Range |
|---|---|
| *Gradient Boosting* | |
| n_estimators | $[50, 2000]$ |
| subsample | $[0.4, 10]$ |
| min_samples_split | $[2, 500]$ |
| learning_rate | $[0.05, 0.5]$ |
| n_iter_no_change | $[5, 100]$ |
| max_leaf_nodes | None |
| max_depth | None |
| *Random Forest* | |
| n_estimators | $[50, 500]$ |
| max_depth | None |
| min_samples_split | $[2, 100]$ |
| max_features | $[0.4, 1.0]$ |
| *Extra Trees* | |
| n_estimators | $[50, 500]$ |
| max_depth | None |
| min_samples_split | $[2, 100]$ |
| max_features | $[0.4, 1.0]$ |
| *Decision Tree* | |
| max_depth | $[1, 2000]$ |
| min_samples_split | $[2, 500]$ |
| min_samples_leaf | $[1, 500]$ |
| max_features | $[0.4, 1.0]$ |

| Hyperparameter | Range |
|---|---|
| *Linear/Logistic Regression* | |
| alpha | $[1e{-}6, 1e6]$ |
| *Gaussian Process* | |
| n_restarts_optimizer | $[0, 5]$ |
| normalize_y | [True, False] |
| alpha | $[1e{-}5, 1e2]$ |
| max_iter_predict | $[100, 1000]$ |
| *Support Vector Machine* | |
| C | $[1e{-}6, 1e6]$ |
| kernel | [linear, poly, rbf, sigmoid] |
| degree | $[1, 10]$ |
| *k-Nearest Neighbors* | |
| n_neighbors | $[1, 30]$ |
| weights | [uniform, distance] |
| *Neural Network* | |
| hidden_layers | $[1, 3]$ |
| hidden_neurons | $[8, 64]$ |
| alpha | $[1e{-}6, 1e1]$ |
| learning_rate_init | $[5e{-}4, 1e{-}2]$ |
| batch_size | $[16, 64]$ |
| activation | [tanh, relu] |
| max_iter | 200 |
| solver | adam |
| early_stopping | True |

Table 5: **Baselearner search spaces.** We describe the search spaces for each baselearner. If a hyperparameter is not mentioned, we use its default. All baselearners are implemented with scikit-learn (Pedregosa et al., 2011); we refer to their documentation for more information.

$i$'s normalized score on objective $j$ as $s_{ij}$. As different risk measures and metrics have different scales, we normalize each of these scores by dividing the raw score $\tilde{s}_{ij}$ with the raw score of a constant ATE baseline $\tilde{s}_j^{\text{ATE}}$: $s_{ij} = \frac{\tilde{s}_{ij}}{\tilde{s}_j^{\text{ATE}}}$.

**Average (normalized) score.** For each pipeline $i$, we compute the normalized average score across objectives:

$$S_i = \frac{1}{m} \sum_{j=1}^{m} s_{ij},$$

with $m$ the number of objectives. We then select the pipeline(s) with the best $S_i$.

**Euclidean distance to the origin.** We compute each pipeline $i$'s Euclidean distance to the origin:

$$D_i = \frac{1}{m} \sqrt{\sum_{j=1}^{m} s_{ij}^2},$$

with $m$ the number of objectives. We then select the pipeline(s) with the lowest $D_i$.

**Average rank.** Rank all pipelines $i$ for each objective $j$, denoted as $r_{ij}$, and compute the average rank:

$$R_i = \frac{1}{m} \sum_{j=1}^{m} r_{ij}.$$

Select the pipeline(s) with the lowest $R_i$.

**Stacking—Constrained Optimization Problem.** To combine multiple pipelines into a stacked estimator, we introduce a procedure that assigns weights $w_{ij}$ (where $0 \leq w_i \leq 1$) to each pipeline $i$, optimizing these weights to minimize the squared error of the weighted prediction with respect to those pseudo-outcomes of objective $j$. We additionally add an $l_2$ regularization term, which can be tuned on a validation set. With multiple objectives, we repeat this for each objective and then average the weights $W_i = \sum_{j=1}^{m} w_{ij}$.

**Stacking—Softmax (Mahajan et al., 2023).** An alternative stacking procedure is to determine the weight of each estimator with a softmax function:

$$w_i j = \frac{\exp(\kappa s_i j)}{\sum_{j=1}^{m} \exp(\kappa s_i k)},$$

with $\kappa$ a temperature parameter that can be tuned. With multiple objectives, we repeat this for each objective and then average the weights $W_i = \sum_{j=1}^{m} w_{ij}$.

**Pareto.** We select all pipelines that are Pareto optimal, meaning no other pipeline $k$ satisfies:

$$s_{kj} \geq s_{ij} \; \forall j \; \text{ and } \; s_{kj} > s_{ij} \text{ for at least one } j.$$

## B.6 AUTOCATE'S API: ADDITIONAL INFORMATION

We give more information on `AutoCATE`'s initialization arguments in Listing 1.

```python
class AutoCATE:
    def __init__(
        self,
        # evaluation_metrics: Risk measures to evaluate the performance
        evaluation_metrics=None,
        # preprocessors: Preprocessors to try (defaults added later)
        preprocessors=None,
        # base_learners: Baselearners to try (defaults added later)
        base_learners=None,
```

```
10          # metalearners: Metalearners to try (defaults added later)
11          metalearners=None,
12          # task: Type of task ('regression' or 'classification')
13          task="regression",
14          # metric: Metric used to evaluate the model (e.g., 'MSE')
15          metric="MSE",
16          # ensemble_strategy: Strategy for selecting a final model
17          ensemble_strategy="top1average",
18          # single_base_learner: Use only one base learner
19          single_base_learner=False,
20          # joint_optimization: Same hyperparameters for baselearners
21          joint_optimization=False,
22          # n_folds: Number of folds for cross-validation
23          n_folds=1,
24          # n_trials: How many trials to optimize the estimation pipeline
25          n_trials=50,
26          # n_eval_versions: Number of versions of each risk measure
27          n_eval_versions=1,
28          # n_eval_trials: Number of trials for evaluating the model
29          n_eval_trials=50,
30          # seed: Random seed for reproducibility
31          seed=42,
32          # visualize: Whether to visualize results
33          visualize=False,
34          # max_time: Maximum time allowed for fitting the model
35          max_time=None,
36          # n_jobs: Number of parallel jobs to run
37          n_jobs=-1,
38          # cross_val_predict_folds: Folds for cross-validated estimates
39          cross_val_predict_folds=1,
40          # holdout_ratio: Ratio of data for validation (if single fold)
41          holdout_ratio=0.3
42      ):
43
44          # Initialization code (not included here)
45          ...
```

Listing 1: **Arguments for the `AutoCATE` class initialization.** We describe each argument and its default initialization.

## C  DATA: ADDITIONAL INFORMATION

This section describes the data used in this work in more detail.

**IHDP (Hill, 2011).**   The data come from the Infant Health and Development Program, describing the impact of child care and home visits on children's cognitive development. Treatments and outcomes were simulated for a total of 100 data sets. Each version contains $n = 747$ instances and $d = 25$ covariates.

**ACIC (Dorie et al., 2019).**   The data from the ACIC 2016 competition was based on data from the Collaborative Perinatal Project, studying drivers of developmental disorders in pregnant women and their children. 77 distinct data sets were created, each with $n = 4,802$ instances and $d = 58$ covariates. 100 iterations were originally created for each data set, but we use only the first one for each.

**Twins (Louizos et al., 2017).**   The Twins data studies the effect of being the heavier twin on mortaility. $n = 11,984$ pairs of twins are included, with $d = 46$ features each. Only one version of this data set exists, so we run 10 iterations of each experiment.

**News (Johansson et al., 2016).**   This data simulates a reader's reading experience ($y$) based on the device they use for reading ($t$) and the news article ($x$). There are 50 distinct data sets, each with $n = 5,000$ instances with and $d = 3,477$ covariates.

Below, we include results for two data sets on uplift modeling:

**Hillstrom (Hillstrom, 2008).** This data contains records of customers ($n = 64{,}000$) that were contacted by a marketing campaign over e-mail. Originally, customers received either no mail, a mail with men's merchandise, or one with women's merchandise, but we convert it to not contacted ($t = 0$) or contacted ($t = 0$). For each customer, $d = 10$ covariates are available. As the outcome $y$, we consider whether the customer visited the website or not.

**Information (Larsen, 2023).** The information data set comes from the R Information package. It describes customers ($n = 10{,}000$, $d = 68$) in the insurance industry, as well as whether they were contacted with a marketing campaign and whether they made a purchase.

## D    ADDITIONAL RESULTS

### D.1    STAGE 1: EVALUATION

Table 6 shows results for evaluating with $k$-fold cross validation for different values of $k$.

|        | 1              | 2              | 3              | 4              | 5              | 10             |
|--------|----------------|----------------|----------------|----------------|----------------|----------------|
| *IHDP* | $2.15_{\pm.35}$ | $2.16_{\pm.35}$ | $2.10_{\pm.35}$ | $2.07_{\pm.33}$ | $2.29_{\pm.42}$ | $2.25_{\pm.41}$ |
| *ACIC* | $1.52_{\pm.09}$ | $1.58_{\pm.08}$ | $1.48_{\pm.08}$ | $1.51_{\pm.09}$ | $1.50_{\pm.08}$ | $1.53_{\pm.09}$ |
| *Twins* | $.323_{\pm.00}$ | $.324_{\pm.00}$ | $.322_{\pm.00}$ | $.324_{\pm.00}$ | $.344_{\pm.00}$ | $.346_{\pm.00}$ |
| *News* | $2.42_{\pm.07}$ | $2.40_{\pm.07}$ | $2.41_{\pm.06}$ | $2.41_{\pm.07}$ | $2.45_{\pm.07}$ | $2.45_{\pm.07}$ |

Table 6: **The effect of $k$ in $k$-fold cross validation.** For each data set, we show result for a varying number of cross-validation folds. Results for 50 evaluation trials with a $T$-risk and 50 estimation trials with a $T$-Learner and gradient boosting.

Risk measures may suffer from congeniality bias, by being predisposed to favor their related metalearners (Curth & van der Schaar, 2023). For example, a $T$-risk may pick a $T$-Learner more often, even when it is suboptimal. The results in our main body found that the $T$-risk works very well with a $T$-Learner, but these results may not hold in general due to congeniality bias. Therefore, we again compare the different risk measures when estimating with either $S$-Learners only or selected metalearners in Table 7

### D.2    STAGE 2: ESTIMATION

Figure 9 shows how often each metalearner gets picked in `AutoCATE`'s BestMeta configuration. The difference in metalearner selection rates illustrates the importance of data-driven metalearner selection, as facilitated by `AutoCATE`. Interestingly, other metalearners are preferred for a binary outcome (Twins) than for continuous outcomes (all others). This finding suggests that different BestMeta configurations may be optimal for different outcomes.

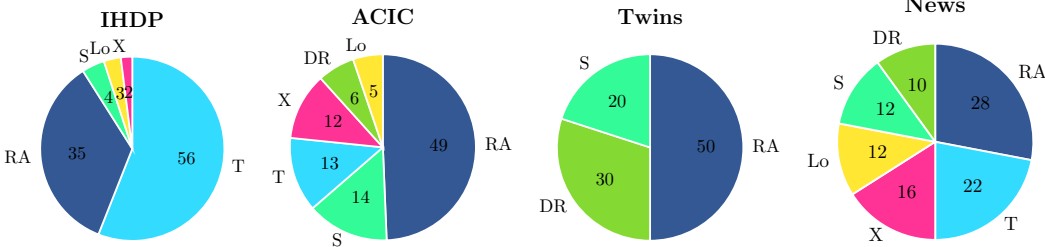

Figure 9: **Metalearner selection.** We show how many times a metalearner gets picked (in % of all data set iterations) for a given data set. Results for `AutoCATE`'s BestMeta configuration, including the $S$-, $T$-, $Lo$-, $X$-, $RA$-, $DR$-, and $U$-Learners, with 50 evaluation and 500 estimation trials.

|  | DR | F | IF | kNN | R | T | U | Z |
|---|---|---|---|---|---|---|---|---|
| *IHDP* | $3.21_{\pm.55}$ | $3.64_{\pm.60}$ | $4.60_{\pm.78}$ | $\underline{3.11}_{\pm.53}$ | $3.48_{\pm.58}$ | $\mathbf{3.10}_{\pm.54}$ | $3.62_{\pm.58}$ | $4.12_{\pm.70}$ |
| *ACIC* | $\underline{1.61}_{\pm.09}$ | $1.79_{\pm.10}$ | $2.07_{\pm.10}$ | $1.88_{\pm.09}$ | $1.73_{\pm.10}$ | $\mathbf{1.58}_{\pm.09}$ | $1.85_{\pm.10}$ | $2.16_{\pm.12}$ |
| *Twins* | $.328_{\pm.00}$ | $.328_{\pm.00}$ | $.347_{\pm.02}$ | $\underline{.320}_{\pm.00}$ | $.325_{\pm.00}$ | $\mathbf{.320}_{\pm.00}$ | $.321_{\pm.00}$ | $.330_{\pm.00}$ |
| *News* | $\underline{2.47}_{\pm.09}$ | $\underline{2.51}_{\pm.08}$ | $2.97_{\pm.13}$ | $\underline{2.49}_{\pm.09}$ | $2.76_{\pm.12}$ | $\mathbf{2.46}_{\pm.08}$ | $2.78_{\pm.13}$ | $2.99_{\pm.14}$ |

(a) Estimation with an $S$-Learner

|  | DR | F | IF | kNN | R | T | U | Z |
|---|---|---|---|---|---|---|---|---|
| *IHDP* | $\mathbf{2.07}_{\pm.32}$ | $3.43_{\pm.60}$ | $5.75_{\pm.70}$ | $\underline{2.11}_{\pm.34}$ | $3.45_{\pm.56}$ | $\underline{2.17}_{\pm.37}$ | $3.18_{\pm.56}$ | $4.38_{\pm.71}$ |
| *ACIC* | $\underline{1.40}_{\pm.09}$ | $1.87_{\pm.11}$ | $2.24_{\pm.14}$ | $1.97_{\pm.13}$ | $1.57_{\pm.10}$ | $\mathbf{1.35}_{\pm.09}$ | $1.79_{\pm.11}$ | $2.16_{\pm.11}$ |
| *Twins* | $.328_{\pm.00}$ | $.327_{\pm.00}$ | $.384_{\pm.03}$ | $\mathbf{.324}_{\pm.00}$ | $.328_{\pm.00}$ | $.326_{\pm.00}$ | $.344_{\pm.01}$ | $.348_{\pm.01}$ |
| *News* | $\mathbf{2.42}_{\pm.07}$ | $2.60_{\pm.08}$ | $2.95_{\pm.12}$ | $\underline{2.42}_{\pm.07}$ | $2.75_{\pm.15}$ | $\underline{2.43}_{\pm.07}$ | $2.78_{\pm.13}$ | $2.77_{\pm.11}$ |

(b) Estimation with selected metalearners (BestMeta configuration: $S$, $T$, $DR$, $X$, $RA$, $Lo$)

Table 7: **Performance for validation based on different risk measures.** Results in $\sqrt{\text{PEHE}}_{\pm\text{SE}}$ (lower is better). **Bold** highlights the best results, with underlined values falling within 1 standard error. Results for 50 evaluation trials and 50 estimation trials with a gradient boosting baselearner.

We compare different metalearners in terms of $\sqrt{\text{PEHE}}$ in Table 8. These results show that searching across metalearners typically significantly improves precision compared to using only one metalearner. Moreover, some metalearners can result in very poor performance even after 200 optimization trials. Typically, these results are due to exceptionally poor performance in some iterations (e.g., the $R$-Learner). Additionally, we compare the performance trade-off in terms of time and precision for best metalearners in Figure 10. These results show that the $S$-, $T$-, and $Lo$-Learner are often the fastest to train and the most precise in terms of $\sqrt{\text{PEHE}}$. These results illustrate the potential of improving `AutoCATE`'s time efficiency by considering these trade-offs. To give a sense of AutoCATE's runtime, we include the required computation times to run `AutoCATE` on different data sets in Table 9. Although some time is required, running our framework locally is feasible for small to moderate data sets.

|  | S | T | DR | X | R | RA | Lo | Z | U | F | AllMeta |
|---|---|---|---|---|---|---|---|---|---|---|---|
| *IHDP* | $4.52_{\pm.74}$ | $2.52_{\pm.37}$ | $5.91_{\pm.98}$ | $5.46_{\pm.87}$ | $2752.36_{\pm1613.91}$ | $5.80_{\pm.89}$ | $2.47_{\pm.34}$ | $50.09_{\pm6.21}$ | $7.45_{\pm1.12}$ | $9.58_{\pm.95}$ | $\mathbf{1.54}_{\pm.25}$ $(-37.5\%)$ |
| *ACIC* | $4.00_{\pm.24}$ | $4.26_{\pm.14}$ | $3.61_{\pm.22}$ | $3.09_{\pm.16}$ | $477325.02_{\pm87957.53}$ | $3.27_{\pm.19}$ | $3.07_{\pm.10}$ | $150829.14_{\pm56790.59}$ | $5.75_{\pm.43}$ | $4.65_{\pm.44}$ | $\mathbf{1.62}_{\pm.09}$ $(-47.3\%)$ |
| *Twins* | $\mathbf{.318}_{\pm.00}$ | $.345_{\pm.01}$ | $.320_{\pm.00}$ | $.333_{\pm.00}$ | $77.408_{\pm33.07}$ | $.323_{\pm.00}$ | $.360_{\pm.00}$ | $.546_{\pm.01}$ | $.418_{\pm.01}$ | $.376_{\pm.00}$ | $.321_{\pm.00}$ $(+\ 0.9\%)$ |
| *News* | $2.89_{\pm.14}$ | $2.53_{\pm.07}$ | $3.38_{\pm.15}$ | $2.93_{\pm.13}$ | $36448.74_{\pm13452.34}$ | $3.14_{\pm.13}$ | $2.57_{\pm.08}$ | $16.06_{\pm1.80}$ | $2.74_{\pm.13}$ | $3.41_{\pm.11}$ | $\mathbf{2.40}_{\pm.08}$ $(-\ 5.0\%)$ |

Table 8: **Comparing metalearner precision.** For each data set, we compare the different metalearner's performance in terms of $\sqrt{\text{PEHE}}$, with the best result highlighted in **bold**. We also include a comparison with searching over all metalearners (AllMeta) and, in brackets, show how much this outperforms the best single metalearner. For each result, `AutoCATE` uses a $T$-risk with 50 evaluation trials, 200 estimation trials, and top 1 average model selection.

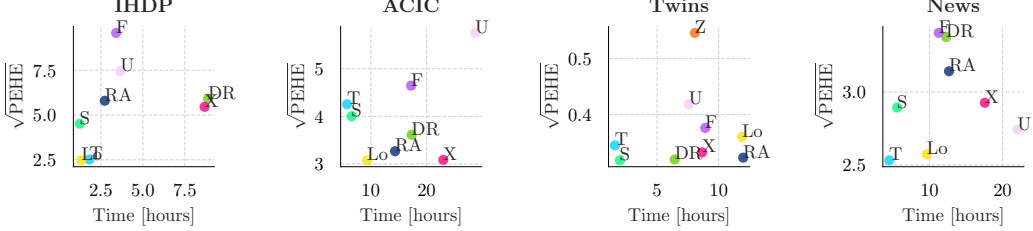

Figure 10: **Comparing metalearner precision and time efficiency.** We show each metalearner's performance in precision ($\sqrt{\text{PEHE}}$) and time (excluding outliers, see Table 8). For each, `AutoCATE` uses a $T$-risk with 50 evaluation trials, 200 estimation trials, and top 1 average model selection.

A key innovation for `AutoCATE` is that it optimizes the entire ML pipeline, including preprocessing steps. In Table 10, we present an ablation study for our framework with and without preprocess-

| **IHDP** | **ACIC** | **Twins** | **News** |
| $n = 747; d = 25$ | $n = 4{,}802; d = 58$ | $n = 11{,}984; d = 46$ | $n = 5{,}000; d = 3{,}477$ |
| 1'21" | 6'00" | 29'38" | 6'49" |

Table 9: **AutoCATE time complexity.** We show the average runtime required to run AutoCATE's complete, end-to-end optimization on a single iteration of different data sets. For each data set, we include the size ($n$) and dimensionality ($d$). AutoCATE uses 50 evaluation trials and 50 estimation trials with the BestMeta–BestBase configuration. These experiments were conducted locally, on a machine with an AMD Ryzen 7 PRO 4750U processor (1.70 GHz), 32 GB of RAM, and a 64-bit operating system.

|  | **Preprocessing** | |
|  | ✓ | ✗ |
|---|---|---|
| *IHDP* | $\mathbf{1.25}_{\pm.18}$ | $1.69_{\pm.27}$ |
| *ACIC* | $\mathbf{1.52}_{\pm.09}$ | $\underline{1.58}_{\pm.09}$ |
| *Twins* | $\mathbf{.315}_{\pm.00}$ | $.320_{\pm.00}$ |
| *News* | $\mathbf{2.33}_{\pm.06}$ | $\underline{2.38}_{\pm.07}$ |

Table 10: **Analayzing the added value of preprocessing.** We compare AutoCATE's performance with and without preprocessing included in the search space, in terms of $\sqrt{\text{PEHE}}$, with the best result highlighted in **bold**. Preprocessing includes feature scaling and selection. AutoCATE results for a $T$-risk with 50 evaluation trials and 50 estimation trials with the BestMeta–BestBase configuration.

ing. For all data sets, AutoCATE achieves the best performance *with* preprocessing, though the improvement is only significant for the IHDP and Twins data.

We can also apply explainability techniques to understand what drives a pipeline's predictions. Figure 11 illustrates this and shows how permutation feature importance can be used with AutoCATE.

### D.3 STAGE 3: ENSEMBLING

The ensemble built by AutoCATE can be used to gauge the uncertainty regarding a prediction, by highlighting the spread of predictions. We illustrate such an analysis in Figure 12.

### D.4 BENCHMARKING AutoCATE

Table 11 presents results for additional benchmarks: S- and T-Learners based on linear or logistic models (without regularization).

Figure 13 shows additional results for two data sets for uplift modeling (see Appendix C for more information on the data). The effectiveness of AutoCATE is related to at least three factors. First, by using the AUQC metric, the search is aligned with the downstream task: prioritizing instances

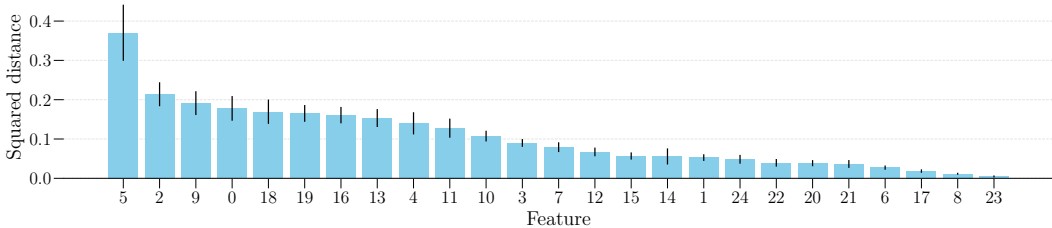

Figure 11: **Analyzing AutoCATE's feature importance.** We can analyze how much each feature contributes to treatment effect heterogeneity. We illustrate this analysis for the first iteration of IHDP using permutation feature importance, showing the squared distance to the original prediction when permuting a feature column.

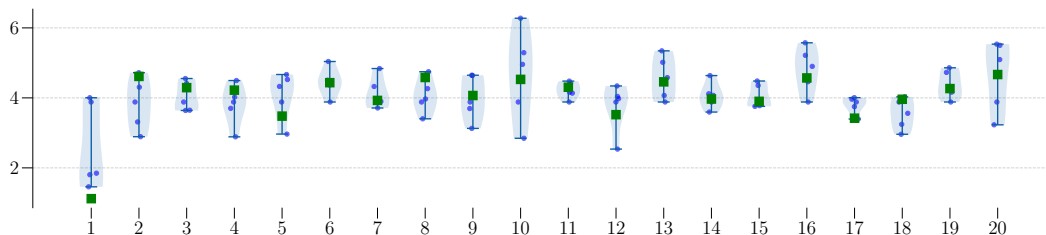

Figure 12: **Assessing uncertainty with `AutoCATE`.** The ensemble returned by `AutoCATE` can be used to analyze uncertainty regarding the prediction. We illustrate this for the first 20 instances of the first iteration of the IHDP data. For each instance, the (usually unknown) ground truth is shown in green, while the predictions from the top five pipelines are shown in blue and with a violinplot.

| | **AutoCATE** | | **Benchmarks** | | | | | |
|---|---|---|---|---|---|---|---|---|
| | *Top 1* | *Top 5* | *S–RF* | *T–RF* | *S–GB* | *T–GB* | *S–LR* | *T–LR* |
| *IHDP* | **1.25**$_{\pm.18}$ | 1.38$_{\pm.21}$ | 3.30$_{\pm.57}$ | 2.61$_{\pm.45}$ | 3.02$_{\pm.52}$ | 1.86$_{\pm.29}$ | 5.73$_{\pm.89}$ | 2.41$_{\pm.39}$ |
| *ACIC* | 1.52$_{\pm.09}$ | 1.45$_{\pm.10}$ | 1.67$_{\pm.08}$ | 1.65$_{\pm.09}$ | 1.48$_{\pm.10}$ | **1.38**$_{\pm.09}$ | 4.13$_{\pm.25}$ | 3.08$_{\pm.15}$ |
| *Twins* | .315$_{\pm.00}$ | **.314**$_{\pm.00}$ | .318$_{\pm.00}$ | .331$_{\pm.00}$ | .319$_{\pm.00}$ | .334$_{\pm.00}$ | .320$_{\pm.00}$ | .335$_{\pm.00}$ |
| *News* | 2.33$_{\pm.06}$ | **2.29**$_{\pm.06}$ | 2.46$_{\pm.09}$ | 2.39$_{\pm.07}$ | 2.68$_{\pm.11}$ | 2.40$_{\pm.06}$ | 3.68$_{\pm.17}$ | 2.93$_{\pm.12}$ |

Table 11: **Comparing `AutoCATE` with common benchmarks on CATE estimation.** We compare performance in terms of $\sqrt{\text{PEHE}}$, with the best result highlighted in **bold**. `AutoCATE` results for a $T$-risk with 50 evaluation trials and 50 estimation trials with the BestMeta–BestBase configuration.

for treatment (Vanderschueren et al., 2024). Second, the search space for AutoCATE includes more meta- and baselearners than the benchmarks. Third, the top five ensemble seems to improve the stability and accuracy of the predicted ranking.

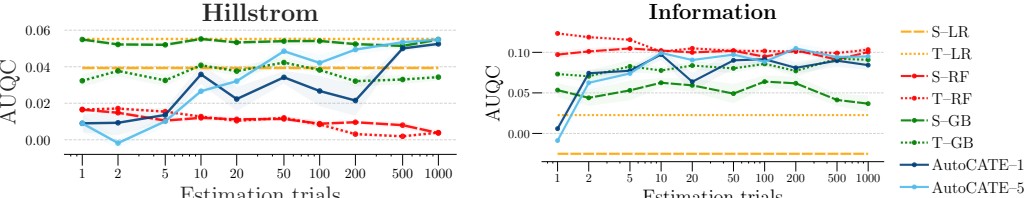

Figure 13: **Benchmarking `AutoCATE` for treatment prioritization.** We present additional results in terms of AUQC for two uplift data sets, Hillstrom and Information. These show that `AutoCATE` is a useful tool for prioritizing instances for treatment, and highlight that its optimization is more effective at optimizing AUQC compared to the benchmarks based on $\mu$-risk. `AutoCATE` uses a $T$-risk with 50 evaluation trials and the AUQC metric, the BestMeta-BestBase search space, and Top 1 or Top 5 ensembling.

### D.5 ANALYZING `AUTOCATE`'S RESULTS

We analyze the results of `AutoCATE`'s optimized pipelines in Figure 14. These results illustrate how `AutoCATE` can facilitate a higher-level, comprehensive analysis of methods for CATE estimation and model validation.

## E COMPARING SOFTWARE PACKAGES FOR CATE ESTIMATION

Table 12 lists software packages for CATE estimation, comparing their functionalities with `AutoCATE`. Notably, *no other package* is focused on automated, end-to-end CATE estimation.

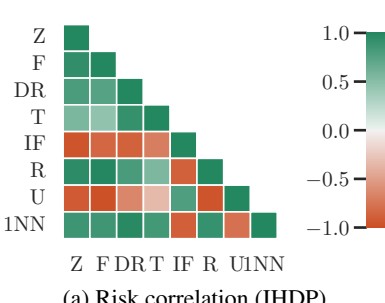
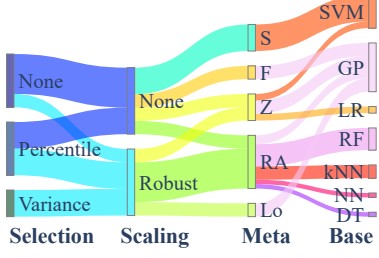

(a) Risk correlation (IHDP)      (b) Pipeline composition (Twins)

Figure 14: **Analyzing `AutoCATE`'s results.** We present results analyzing pipelines optimized by `AutoCATE`. Figure (a) shows the correlation between risk measures for a single IHDP iteration. Surprisingly, risk measures can be strongly *negatively correlated*, suggesting potential for more advanced multi-objective approaches that adaptively learn which objectives are reliable for a given data set. Figure (b) visualizes the *optimal pipelines* learned across ten iterations for the Twins data.

| PACKAGE Name | FUNCTIONALITIES (1) | (2) | (3) | (4) | GENERAL INFORMATION Language | Reference | Link |
|---|---|---|---|---|---|---|---|
| CausalML | ✗[*] | ✓ | ✗ | ✗ | Python | Chen et al. (2020) | GitHub |
| EconML | ✓[§] | ✓ | ✓[§] | ✗ | Python | — | GitHub |
| DoWhy | ✗[†] | ✓ | ✗ | ✗ | Python | Sharma & Kiciman (2020) | GitHub |
| Causica | ✗ | ✓ | ✗ | ✗ | Python | Geffner et al. (2022) | GitHub |
| UpliftML | ✗ | ✓ | ✗ | ✗ | Python | Teinemaa et al. (2021) | GitHub |
| scikit-uplift | ✗ | ✗ | ✗ | ✗ | Python | — | GitHub |
| grf | ✗ | ✓ | ✓[‡] | ✗ | R | Wager & Athey (2018) | CRAN |
| `AutoCATE` | ✓ | ✓ | ✓ | ✓ | Python | This work | GitHub |

[*]CausalML offers provides some tools for internal validity, such as comparing results across segments.
[§]EconML includes an $R$-risk and can provide an ensemble based on this risk measure.
[†]DoWhy includes robustness checks for assumption violations.
[‡]The grf package allows for evaluation based on the Targeting Operating Characteristics curve.

Table 12: **Software package comparison.** We provide an overview of commonly used packages for CATE estimation and compare their functionalities with `AutoCATE`, showing whether they support (1) evaluation, (2) estimation, (3) ensembling, and (4) automated, end-to-end optimization—as provided by `AutoCATE` or similar.

