# OpenReview forum: "AutoCATE: End-to-End, Automated Treatment Effect Estimation"
_ICLR.cc/2025/Conference — Submitted to ICLR 2025_

### Official Review · Reviewer_ztLg · 2024-10-31

**Soundness:** 2
**Presentation:** 1
**Contribution:** 1
**Rating:** 1
**Confidence:** 4

**Summary:**

Accurate estimation of heterogeneous treatment effects is critical in domains such as healthcare, economics, and education. Real-world adoption of these methods in estimating conditional average treatment effects (CATE) remains limited due to the complexity of implementation, tuning, and validation. To address the challenges, the authors aim to develop ML pipelines for CATE estimation through automated, end-to-end protocols. The authors introduce a model termed AutoCATE, which is designed to find automated solutions tailored for CATE estimation based on protocols for evaluation, estimation, and ensembling. The automated solution is obtained after formalizing the potential problem as a counterfactual Combined Algorithm Selection and Hyperparameter optimization (CASH) problem. The authors also undergo comprehensive experiments and demonstrate that the results obtained by the proposed model outperform the results obtained by other common strategies.

**Strengths:**

As pointed out by the authors, this paper proposes a practical and comprehensive solution as the automated, end-to-end construction and validation of ML pipelines for CATE estimation.

**Weaknesses:**

It seems that the paper is simply a user manual for AutoCATE in Python. The authors do not provide any new models/methodologies to estimate CATE. All the methods in the paper are presented in previous papers. One can simply follow the algorithms given in existing literature and replicate the result. Typically, the paper does not contribute much to the related society.

Another clear drawback is poor writing. Many notational mistakes appear in the appendix. For example, see line 855. Furthermore, authors only outline the general procedure without providing a detailed description/explanation, which causes much confusion among readers. For instance, the authors state that, in the evaluation stage, the proposed model would combine different measures to make the evaluation more robust (see line 212).  Nevertheless, the authors do not explain how to combine different measures. The combination may simply concatenate different measures to form a vector or add all the measures together. Statements with unclear descriptions and explanations reduce the readability of the paper.

**Questions:**

No technical questions. Please see the weakness above.

---

> ### Author Response · Authors · 2024-11-16
> **Response to reviewer ztLg [Part 1/3]**
>
> Thank you for reviewing our work! Below, we respond to both points mentioned in your review: (1) novelty and significance of our contributions, and (2) writing.
>
> ___
>
> ## _Novelty and significance of our contributions_
>
> First, we would like to clarify the problem we address and more clearly describe the unique contributions of our work.
>
> **Problem statement**
>
> Recent years have seen significant advances in using ML for CATE estimation. However, many questions remain regarding the application of these methods in practice, which hampers their real-world adoption.
> First, many methods have been proposed for CATE estimation, but we do not yet clearly enough understand _when_ what methods are preferable. It even remains an open question how to best tune these methods, as no established guidelines exist. Besides modelling, aspects such as preprocessing or ensembling have received almost no consideration in previous work.
> Second, while there has been recent progress on understanding model selection in this context, different studies use vastly different setups and come to different conclusions–meaning that there is no consensus as to which approach works best.
>
> **Our solution**
>
> We argue that a _different perspective_ is needed to tackle these questions and bridge the gap between theoretical advancements in CATE estimation and their real-world application. We focus on automatically building and finding an optimal ML pipeline for CATE estimation: our proposed solution, AutoCATE, is the first approach that can tackle this end-to-end and in an automated manner. Both these characteristics are _important_ and _novel_, representing significant advances.
> Adopting an **end-to-end** approach enables us to consider _all relevant design choices_–such as preprocessing, model selection, and ensembling–together, rather than in isolation.
> Our emphasis on **automation** ensures the replicability and practicality of our protocols. This way, our final configuration of AutoCATE is a **single algorithm** for finding an optimal ML pipeline that gives state-of-the-art predictions across a variety of settings.
>
> **Methodological contributions**
>
> Developing this solution requires overcoming **significant challenges**. While individual components of our approach appear in prior work, a core contribution of AutoCATE lies in integrating these components into a fully automated, end-to-end framework. For example, while previous work focuses on comparing risk measures for model selection, we additionally analyze how to construct the ML pipelines underlying these risk measures. This is a clear consideration in practice, which had not yet been tackled in prior work. Finally, through this automated, end-to-end perspective, we uncover novel solutions to critical issues, such as how to effectively split data for training and validation or how to perform multi-objective optimization with multiple risk measures.
>
> As such, AutoCATE makes **several distinct and important contributions**. Methodologically, we provide the first unified algorithm that automates all stages of CATE estimation, from preprocessing to estimation, model selection and ensembling. Some functionalities of our framework are even _novel methodologically_, such as the multi-objective optimization, or re-discover methods that have been largely abandoned, such as the Lo-Learner.

---

> ### Author Response · Authors · 2024-11-16
> **Response to reviewer ztLg [Part 2/3]**
>
> **Empirical contributions**
>
> In addition to its methodological contributions, AutoCATE makes significant *empirical contributions*. Prior studies examine questions related to one of the three stages considered in AutoCATE in isolation. Conversely, our more holistic experimental setup provides more practical conclusions that lead directly to implementable solutions within our AutoCATE framework, which marks a substantial shift toward actionable guidance for both researchers and practitioners. This way, we can progress beyond earlier findings by revisiting similar questions in a more practical context enabled by automation. Our analyses uncover novel insights, challenging the established knowledge in CATE estimation. For example, Figure 1 shows that vastly different metalearners can be optimal for a given data generating process and highlights our limited understanding of this problem.
>
> Other fields have long benefited from meta-analyses and systematic comparisons, enabling more refined insights into model performance across scenarios. However, a similar level of synthesis and clarity is currently _missing_ in ML for CATE estimation, underscoring a critical need for comprehensive, end-to-end solutions that address these challenges. Through our framework, we conduct arguably the **most comprehensive analysis to date** of methods for both CATE estimation and model selection. We evaluate a wide array of data sets spanning classification and regression outcomes, varying in size, dimensionality, and data-generating processes. Similarly, we investigate a vastly more diverse collection of methods for CATE estimation and model selection, exploring their interplay in an automated and standardized framework. We emphasize that “benchmarks” are clearly included in ICLR’s call for papers.
>
> **Engineering contribution**
>
> A final contribution of our work is making AutoCATE accessible as an open-source software package. Our framework allows researchers to compare methods more easily, more fairly (based on similar tuning procedures), and more holistically (e.g., by analyzing whether a novel method is complementary), thereby supporting future innovations in this field. The low-code API will democratize access to these advanced methods by ensuring accessibility for practitioners unfamiliar with machine learning. Although our work is not limited to this package, we would like to emphasize that software can constitute an important research contribution. “Software libraries” is explicitly included as a topic in ICLR’s call for papers and many recent ICLR publications offer such contributions (e.g., Schneider, Balles, & Hennig, 2019; Dangel, Kunstner, & Hennig, 2020; Jarrett et al. 2021; Jiminez et al., 2024; Hvarfner, Hutter, & Nardi, 2024).
>
> - **Action taken:** We have revised our paper to more clearly stress the addressed research gaps with respect to prior work and describe the contributions of our work.
>
> ## Writing
>
> We have revised the writing and proofread our notation again. Respond to the issues raised in your review specifically:
> Thank you for catching that notational error! This has now been corrected.
> The combination of risk measures was introduced in sections 4.3 and 5.4. However, we appreciate that this explanation may not have provided enough detail. To address this, we have added a more detailed description in the appendix, where we explain the various ways in which the risk measures can be combined and describe the ensembling approaches explored in our work.
>
> If the reviewer has any remaining concerns or if there are other areas requiring further clarification, we would be happy to revise the paper accordingly.
>
> ___
>
> We hope that these answers address your concerns. If not, please feel free to let us know, and we would be happy to provide further information.

---

> > ### Author Response · Authors · 2024-11-16
> > **Response to reviewer ztLg [Part 3/3]**
> >
> > ___
> >
> > ### References:
> > - Schneider, F., Balles, L., & Hennig, P. DeepOBS: A Deep Learning Optimizer Benchmark Suite. (2019). In International Conference on Learning Representations.
> > - Dangel, F., Kunstner, F., & Hennig, P. (2020). BackPACK: Packing more into Backprop. In International Conference on Learning Representations.
> > - Jarrett, D., Yoon, J., Bica, I., Qian, Z., Ercole, A., & van der Schaar, M. (2021). Clairvoyance: A Pipeline Toolkit for Medical Time Series. In International Conference on Learning Representations.
> > - Jimenez, Carlos E., John Yang, Alexander Wettig, Shunyu Yao, Kexin Pei, Ofir Press, and Karthik R. Narasimhan. (2024). "SWE-bench: Can Language Models Resolve Real-world Github Issues?." In The Twelfth International Conference on Learning Representations.
> > - Kahl, Kim-Celine, Carsten T. Lüth, Maximilian Zenk, Klaus Maier-Hein, and Paul F. Jaeger. (2024). "ValUES: A Framework for Systematic Validation of Uncertainty Estimation in Semantic Segmentation." In The Twelfth International Conference on Learning Representations.
> > - Hvarfner, C., Hutter, F., & Nardi, L. (2024). A General Framework for User-Guided Bayesian Optimization. In The Twelfth International Conference on Learning Representations.

---

> > > ### Author Response · Authors · 2024-11-24
> > >
> > > Dear Reviewer,
> > >
> > > Thank you again for the time and effort you’ve dedicated to reviewing our submission!
> > >
> > > As the rebuttal period is drawing to a close, we wanted to kindly check if our responses have addressed your concerns. If so, we would greatly appreciate it if you could consider updating your score accordingly.
> > >
> > > Of course, we remain happy to engage further or clarify any additional questions or comments you may have.
> > >
> > > Thank you again for your thoughtful feedback and support during this process!
> > >
> > > Best regards,
> > > Authors of Submission 3712

---

> ### Comment · Reviewer_ztLg · 2024-11-26
>
> Confused notations: It is strange that minimization is taken over on $a$ and $h$, but the loss function does not contain $a$ and $h$. It is hard for readers to understand at the first glance. Should it be better to introduce a new notational system such that readers can understand directly? See other notations errors in Appendix B (X-learner), $f_x^1$ is not given, or it should be $f_X^1$.
>
> Honestly, we can run different models independently and choose the best. What is the difference between applying the proposed model and choosing the best after running models independently? Please provide justifications. According to the replies, I also concerned if authors include ALL the methods of computing CATE. Missing any one method can make the result/conclusion unreliable.
>
> I am still concerned with the contributions of this work since this is mainly a proof of concept paper, therefore I will keep my score.

---

> > ### Author Response · Authors · 2024-11-27
> > **Response to Reviewer ztLg [Part 1/3]**
> >
> > Thank you for your response and for engaging with us during the rebuttal! Below, we address each of your points concerning notation and the need for automated approaches.
> > ___
> >
> > ## **Notation**
> >
> > Regarding the notation in the problem formulation, we follow the conventions used in AutoML (e.g., Hutter, Kotthoff, & Vanschoren, 2019). The loss function $\mathcal{L}(a_h | \mathcal{D}_\text{test})$ represents the performance of an ML algorithm for CATE estimation $a$ with hyperparameters $h$. The specific form of the loss function depends on the task, but it is ultimately a function of the CATE estimates, which, in turn, depend on the learning algorithm $a_h$. To clarify this relationship, we could rewrite the notation as $\hat{\tau}(x|a_h)$. We would be happy to make this adjustment if the reviewer prefers.
> >
> > Regarding the X-Learner, this should indeed be a large X, as in $f_X(x) = g(x) f^0_X + (1 − g(x)) f^1_X$. Although we can unfortunately no longer upload a revised pdf, we have fixed this typo. Thank you for pointing this out!
> >
> > ## **The need for automated approaches**
> > We agree with the reviewer that a practitioner could, in principle, construct different models manually and select the best one. However, there are (1) many general advantages to using AutoML, and (2) specific motivations for applying it in the context of CATE estimation–we explain both in more detail below. Accordingly, our package is not intended to replace tools like EconML but to provide complementary functionality.
> > ### **(1) The importance of automated machine learning in general**
> >
> >
> > First, our work is inspired by the challenges automated machine learning aims to address in general. Automating these processes makes ML algorithms more accessible and user-friendly for practitioners with limited expertise in machine learning, while ensuring adherence to best practices embedded in the automated workflow. Moreover, AutoML can incorporate critical aspects of ML pipelines that are often overlooked, such as feature selection and scaling, which are essential for practical applications.
> >
> >
> > Additionally, formalizing the search process enables the use of more efficient strategies, such as Bayesian optimization, to accelerate the discovery of an optimal pipeline. This process is also guided by robust model validation, ensuring that the resulting ML pipelines are reliable and trustworthy. The entire automated algorithm can be validated across diverse datasets and seamlessly applied to new datasets using the same configuration, ensuring more reliable performance in practice.
> >
> >
> > To draw a parallel from classification, while packages like scikit-learn provide foundational tools, AutoML frameworks such as auto-sklearn build on top of them to automate and enhance the workflow. Similarly, our goal with AutoCATE is to create an automated framework that can build upon tools like EconML, which are designed to address different aspects of the problem.

---

> > > ### Author Response · Authors · 2024-11-27
> > > **Response to Reviewer ztLg [Part 2/3]**
> > >
> > > ### **(2) The importance of AutoML for CATE specifically**
> > >
> > >
> > > Automated approaches have been developed for various tasks, including regression, classification, computer vision, and time series forecasting. However, no dedicated solutions currently address the unique challenges of CATE estimation. We argue that automation is particularly crucial for CATE estimation due to the added complexity associated with (1) evaluation and (2) the design of pipelines specifically tailored to CATE estimation.
> > >
> > >
> > > First, simply “selecting the best model” is particularly challenging for CATE estimation. Since the true treatment effects are unknown, we have to rely on evaluation criteria that themselves rely on estimating parameters, necessitating careful optimization and tuning of the underlying models. While existing work, such as Mahajan et al. (2023), provides insights into which criteria perform best and emphasizes the importance of precise model tuning, it does not offer practical procedures for achieving this. In contrast, AutoCATE introduces the first fully automated, end-to-end solution that constructs models for *both* evaluation and estimation.
> > >
> > >
> > > Second, CATE estimation pipelines are significantly more complex than standard supervised learning workflows. Even for experienced data scientists, causal methods introduce additional layers of complexity, limiting their adoption in practice. For instance, a DR-Learner involves up to four distinct ML base-learners: a propensity score model, a control outcome model, a treatment outcome model, and the final effect model. Each of these components can use different ML algorithms, each with its own hyperparameters. This leads to a vast number of potential CATE estimation pipelines to explore. AutoCATE, for example, encompasses 2,187 possible pipelines for CATE estimation *even before considering hyperparameters* (see Appendix B.3). Given this expansive search space, an exhaustive grid search is practically infeasible. Because of this, automated approaches for searching an optimal pipeline are essential in practice.
> > >
> > >
> > > Finally, AutoCATE automates preprocessing steps, such as feature selection and scaling, which leads to improved performance (see Table 10 in the updated paper). While existing CATE estimation packages typically lack such functionality, preprocessing for CATE estimation must also address its unique challenges. For example, a feature that is highly predictive of the outcome, may not be predictive of the treatment effect. By including these steps in our automation, we can guarantee that these preprocessing steps are also finetuned correctly and optimized for the task at hand: accurate CATE estimation.
> > >
> > > ### **Including all methods**
> > >
> > > We are not entirely sure we fully understand the following comment: *“[...] I am also concerned if the authors include ALL the methods of computing CATE. Missing any one method can make the result/conclusion unreliable.”* If the reviewer is concerned that AutoCATE may not include all methods for CATE estimation, we provide a response below. If this does not fully address your concern, we apologize and would be happy to respond more directly if you could kindly clarify your question in more detail.
> > >
> > > One of the key goals of AutoCATE is to include a broad range of methods for evaluation, estimation, and ensembling. For instance, to the best of our knowledge, we incorporate all metalearners that have been proposed in the literature so far. Furthermore, we plan to make it easy for researchers and practitioners to add their own methods through an API in a future version of our package. We envision AutoCATE as a tool that will continue to evolve alongside advancements in the field. Similarly, just as tools like auto-sklearn evolve with the progress in machine learning for classification and regression, we expect AutoCATE to adapt as CATE estimation methods advance.
> > >
> > > Nevertheless, not including every possible ML method for estimating CATE does not mean our conclusions are unreliable. Our primary aim was to introduce and validate a method for automatically searching for a well-performing ML pipeline for CATE estimation. The search space included in AutoCATE is already very extensive, encompassing thousands of potential pipelines. While this search space could be expanded further, we are confident that the main insights we present would remain valid even with additional methods included.

---

> > > > ### Author Response · Authors · 2024-11-27
> > > > **Response to Reviewer ztLg [Part 3/3]**
> > > >
> > > > ## **Proof of concept**
> > > >
> > > > Finally, we are unsure if we fully understand your concern about our work being *“mainly a proof of concept paper”*. While it is true that our work can be seen as a proof of concept, where we introduce a novel approach and demonstrate its feasibility, we firmly believe that this also constitutes an *important scientific contribution*. As we explained in our initial rebuttal and earlier in this response, there is a pressing need for AutoML solutions specifically tailored to CATE estimation, and AutoCATE addresses this gap. We validate our approach with an extensive empirical analysis involving 247 data sets. If the reviewer feels that additional experiments are necessary to further demonstrate our method’s usefulness, we would be more than happy to consider including them.
> > > > ___
> > > >
> > > > Thank you once again for reviewing our work. Please let us know if these additional clarifications address your concerns. If any questions remain, we would be happy to provide further clarification if the reviewer could share their concerns in more detail.
> > > >
> > > > ___
> > > >
> > > > ### References
> > > >
> > > > - Hutter, F., Kotthoff, L., & Vanschoren, J. (2019). Automated machine learning: methods, systems, challenges (p. 219). Springer Nature.

---

> > > > > ### Author Response · Authors · 2024-11-29
> > > > >
> > > > > Dear Reviewer,
> > > > >
> > > > > We hope this message finds you well and wish you a very Happy Thanksgiving!
> > > > >
> > > > > We wanted to kindly follow up to see if our responses have helped address the concerns you raised regarding notation, the included methods, and our contributions. If there are any further clarifications needed or additional points you would like us to address, we are more than happy to provide them. If you feel our responses have resolved your concerns, we hope you might consider revisiting your score.
> > > > >
> > > > > Thank you again for your time and valuable insights throughout this process—they mean a great deal to us!
> > > > >
> > > > > Best regards,
> > > > >
> > > > > The Authors of Submission 3712

---

> ### Author Response · Authors · 2024-12-02
>
> Dear Reviewer,
>
> We hope you are doing well! As the rebuttal period is now coming to an end, we wanted to send a final reminder to kindly ask if our responses have addressed your concerns. If so, we would be deeply grateful if you could consider updating your score, as three other reviewers have already done.
>
> Thank you again for your time and thoughtful feedback!
>
> Best regards,
>
> The Authors of Submission 3712

---

### Official Review · Reviewer_M4nr · 2024-11-03

**Soundness:** 2
**Presentation:** 2
**Contribution:** 2
**Rating:** 5
**Confidence:** 3

**Summary:**

This paper develops an automated, end-to-end protocols for CATE estimation. It applies the techniques in AutoML to addresses complexities in real-world CATE estimation task by automating the process across evaluation, estimation, and ensembling stages. The authors use the experiments to compare the automated strategies in AutoCATE and benchmarking its performance with alternatives.

**Strengths:**

1. This paper presents an automated solution for CATE estimation, effectively consolidating various established methods into a single framework.
2. The AutoCATE code is simple, user-friendly, and accessible to practitioners with limited machine learning expertise.
3. The paper provides a comprehensive analysis of existing CATE methods, supported by extensive experimentation.

**Weaknesses:**

1. The contribution of AutoCATE could be more clearly communicated, possibly due to the writing style. For example:

     --Table 9 compares AutoCATE with alternative CATE estimation packages, highlighting similarities in functionality with EconML. The authors claim that AutoCATE provides "automated, end-to-end optimization," which EconML does not. This distinction could be elaborated more clearly in the main text, and Table 9 should be moved to the main sections to clarify AutoCATE's contribution.

     -- In lines 80-81, the authors state that their approach "addresses key aspects often overlooked in CATE estimation, such as preprocessing, feature selection, or ensembling." However, it’s not evident how these three aspects contribute to AutoCATE's performance throughout the paper. Explicitly demonstrating the impact of these components on performance would strengthen the narrative.

2. The advantages of AutoCATE over existing methods are unclear. Benchmarking AutoCATE against common alternatives should be one of the central experiments, yet:

      -- Only two learners (S- and T-learners) are included. The table 8 presents results for additional benchmarks. I suggest including additional learners from Section B.1 (such as the DR-learner and R-learner). Also, due to similarity between AutoCATE and EconML, including the benchmarking between them may also be helpful.

      -- The reason for AutoCATE's superior performance over these learners is not well-explained. I suggest providing a detailed explanation for why AutoCATE outperforms the S- and T-learner.

3. To me, the main contribution of this paper is the development of a new package that facilitates benchmarking and selection of the best pipeline from existing CATE estimation methods, offering useful choices for users. However, similar benchmarking work was previously conducted by Curth & van der Schaar (2023), and this framework, while practical, does not introduce new algorithmic innovations or significantly advance CATE estimation methods to the level typically expected for ICLR.

**Questions:**

What does the number of evaluation and estimation trials mean? This is not clear to me while reading the paper.

---

> ### Author Response · Authors · 2024-11-16
> **Response to reviewer M4nr [Part 1/4]**
>
> Thank you for your feedback! We address each point raised in your review below.
>
> ## Contributions and novelty
>
> First, allow us to clarify the contributions of our work and address potential misconceptions about its scope. Importantly, while the automated, end-to-end nature of AutoCATE is key, our contribution is not limited to the development of a software package; rather, it represents a significant research effort addressing critical gaps in the literature on CATE estimation.
>
> ### Research gap and novelty with respect to prior work
>
> Recent advances in ML for CATE estimation have been significant, but practical application remains challenging and real-world adoption limited. While many methods exist, it is still unclear when each is preferable, and no established guidelines address their tuning. Key aspects like preprocessing and ensembling have not yet sufficiently been addressed. Furthermore, although progress has been made in model selection, there is no consensus on the best approach. We argue that a _different perspective_ is needed to tackle these questions and bridge the gap between theoretical advancements in CATE estimation and their real-world application.
>
> Earlier work–such as Curth & van der Schaar (2023)--is more limited in scope, aiming to benchmark validation criteria for model selection. Our work addresses a related but distinct problem: the automated, end-to-end learning of an optimal ML pipeline for CATE estimation. This involves optimizing ML pipelines not only for constructing risk measures but also for CATE estimation itself, unifying these stages within a single framework. We explicitly frame this as solving a counterfactual Combined Algorithm Selection and Hyperparameter (CASH) optimization problem, highlighting that we tackle a broader _search problem_ that extends beyond model selection as considered in prior work. Our goal is to obtain an optimized pipeline or ensemble for CATE estimation through an automated and end-to-end approach.
>
> ### Methodological contributions
>
> Our proposed solution, AutoCATE, is the first method for tackling this in an end-to-end and in an automated manner. Both these characteristics represent  _important_ and _novel_ methodological advances.
> Adopting an **end-to-end** approach enables us to consider _all relevant design choices_–such as preprocessing, model selection, and ensembling–together, rather than in isolation.
> Our emphasis on **automation** ensures the replicability and practicality of our protocols. This way, our final configuration of AutoCATE is a **single algorithm** for finding an optimal ML pipeline that gives state-of-the-art predictions across a variety of settings.
>
>
> In developing AutoCATE, we make several important and novel methodological contributions related to different stages in our framework, such as multi-objective search strategies and novel ensembling approaches. Additionally, we address novel questions, such as how to divide data between training and validation.
>
> As noted by the reviewer, another benefit of our holistic perspective is the inclusion of aspects largely overlooked in prior work, such as preprocessing or ensembling. We agree that illustrating how these aspects contribute to AutoCATE's performance would complement our work:
> We provide additional results for AutoCATE without preprocessing (feature scaling and selection) below:
> ### Analyzing the added value of preprocessing
> |         | **Preprocessing** |                |
> |--------:|:-----------------:|:--------------:|
> |         |       _Yes_       |      _No_      |
> |  _IHDP_ |   1.25 $\pm$ .18  | 1.69 $\pm$ .27 |
> |  _ACIC_ |   1.52 $\pm$ .09  | 1.58 $\pm$ .09 |
> | _Twins_ |   .315 $\pm$ .00  | .320 $\pm$ .00 |
> |  _News_ |   2.33 $\pm$ .06  | 2.38 $\pm$ .07 |
>
> Additionally, Figure 14b shows how often preprocessing is included in the final pipeline for the Twins data, illustrating that both feature selection and scaling are included in the optimal pipeline in a majority of the cases.
>
> Regarding ensembling, we analyzed the value of using an ensemble in Tables 3 (a) and (b), as well as Figure 8. As evidenced in Figure 8, there are clear benefits to selecting the best five instead of the single best model.
>
> - **Action taken:** We have added these experimental results and stressed the importance of these components more clearly the revised paper. Thank you for this suggestion!

---

> ### Author Response · Authors · 2024-11-16
> **Response to reviewer M4nr [Part 2/4]**
>
> ### Empirical contributions
>
> Another key contribution of AutoCATE is **empirical**. While Curth & van der Schaar (2023) provided valuable benchmarking, they focus exclusively on model selection, and leave _several questions unanswered_. For example, what is the importance of tuning ML models underlying risk measures? How should this be approached? How should data be split between training and validation? Conversely, we aim to answer these questions with a (1) more holistic framework unifying the three key stages–evaluation, estimation, and ensembling and (2) through an automated approach for tuning the ML pipelines required in these stages.
>
> Our setup provides more practical conclusions that lead directly to implementable solutions within our AutoCATE framework, which marks a substantial shift toward actionable guidance for both researchers and practitioners. This way, we can progress beyond earlier findings by revisiting similar questions in a more practical context enabled by automation. Our analyses uncover novel insights, challenging the established knowledge in CATE estimation. For example, Figure 1 shows that vastly different metalearners can be optimal for a given data generating process and highlights our limited understanding of this problem.
>
> ICLR explicitly includes “benchmarks” in its call for papers. Other areas in ML have long benefited from meta-analyses and systematic comparisons, enabling more refined insights into model performance and design choices across different scenarios. We would argue that this level of synthesis and clarity is currently _missing_ in ML for CATE estimation, underscoring a critical need for in-depth analyses that address these challenges. Our work fills this gap by conducting arguably the _most comprehensive empirical analysis to date_ of methods for both CATE estimation and model selection. We evaluate AutoCATE for a wide array of data sets spanning classification and regression outcomes, varying in size, dimensionality, and data-generating processes. Similarly, we include a more diverse collection of methods for CATE estimation and model selection, exploring their interplay in an automated and standardized framework.
>
> In conclusion, our work makes important empirical contributions, through a systematic meta-analysis, the broad exploration of method interactions, and the practical, actionable insights derived from an automated and holistic framework. Finally, we also hope that AutoCATE enables the community to explore similar questions and deepen our understanding of these methods.
>
> ### Engineering contribution
>
> Finally, as noted by the reviewer, a final significant contribution of our work is development of a new package, made accessible as an open-source software package. Our framework allows researchers to compare methods more easily, more fairly (based on similar tuning procedures), and more holistically (e.g., by analyzing whether a novel method is complementary), thereby supporting future innovations in this field. The low-code API will democratize access to these advanced methods by ensuring accessibility for practitioners unfamiliar with machine learning. Although our work is not limited to this package, we would like to emphasize that software can constitute an important research contribution. “Software libraries” is explicitly included as a topic in ICLR’s call for papers and many recent ICLR publications offer such contributions (e.g., Schneider, Balles, & Hennig, 2019; Dangel, Kunstner, & Hennig, 2020; Jarrett et al. 2021; Jiminez et al., 2024; Hvarfner, Hutter, & Nardi, 2024).
>
> - **Action taken:** We have revised our paper to more clearly stress the addressed research gaps with respect to prior work and describe the contributions of our work.

---

> > ### Author Response · Authors · 2024-11-16
> > **Response to reviewer M4nr [Part 3/4]**
> >
> > ## Benchmarking
> >
> > Thank you for the suggestions! We address each point in turn.
> >
> > _“Only two learners (S- and T-learners) are included”_
> >
> > Allow us to clarify the goal regarding the benchmarks and comparisons of AutoCATE. First, AutoCATE is a method for automatically identify the optimal pipeline for CATE estimation, not a standalone CATE estimator. As such, the S- and T-Learner in itself are not considered as benchmarks to AutoCATE. Instead, the competing approaches are S- and T-Learners combined with random tuning based on the observed MSE risk—an approach commonly used in practice but not incorporated in AutoCATE.
> >
> > While we are open to included additional benchmarks, we want to emphasize that the metalearners suggested by the reviewer (such as the DR- and R-Learner) are _already included_ in the search space of AutoCATE. Therefore, these would not represent benchmarks, but rather ablations of AutoCATE, as the search space can easily be constrained to only considered DR-Learners for example.
> >
> > Additionally, we want to point out that tuning a DR- or R-Learner based on observed MSE is not possible, as they directly predict the treatment effect instead of the observed outcomes. For these metalearners, no established tuning procedures exist–which is an important motivation for our work.
> >
> > If, given this information, the reviewer still believes that specific ablations or benchmarks would strengthen our findings, we would be happy to include them.
> >
> > _Comparing with EconML_
> >
> > While we acknowledge the similarity between AutoCATE and EconML, we believe that EconML and AutoCATE serve complementary, but distinct purposes rather than being competitors. EconML offers a collection of methodologies for CATE estimation, evaluation, and even ensembling. However, it does not provide tools for automatically tuning the models or selecting the best pipeline based on risk measures. As such, a direct apples-to-apples comparison between the two is not feasible. Rather, we see EconML as offering methodologies that could be integrated as building blocks for future versions AutoCATE, enriching its functionalities within the automated setup.
> >
> > _“The reason for AutoCATE's superior performance over these learners is not well-explained.”_
> >
> > We appreciate the reviewer’s feedback and acknowledge that this may not have been sufficiently clear. AutoCATE’s improved performance related to the benchmarks are due to two factors:
> > AutoCATE’s **search space** is larger and contains a variety of metalearners and baselearners, including the ones considered by the benchmarks. This additional flexibility is likely to result in better performance by finding the most appropriate modeling pipeline, given that the search continues for long enough.
> > Additionally, AutoCATE includes automated tuning of the preprocessing algorithms in its ML pipelines, which is not included in the benchmarks.
> > AutoCATE’s **model selection** is guided by a T-risk measure, which more accurately reflects the actual objective of CATE estimation (i.e., minimizing the treatment effect prediction error) compared to the MSE in predicting the observed outcome.
> >
> > - **Action taken:** We have included a more detailed explanation of these factors in our updated paper.
> >
> > ## “What does the number of evaluation and estimation trials mean?”
> >
> > To clarify, the number of evaluation and estimation trials refers to the number of distinct pipeline configurations that AutoCATE evaluates and tunes during its search process.
> > 50 **evaluation trials** means that for each type of model underlying the risk measure, AutoCATE constructs and evaluates 50 different machine learning pipelines.
> > 50 **estimation trials** means that AutoCATE builds and compares 50 distinct machine learning pipelines for CATE estimation, each consisting of different combinations of metalearners and baselearners.
> >
> > - **Action taken**: We have revised the text to make this more clear in the paper. Thank you for bringing this to our attention!
> > ___
> >
> > Thank you, once again, for reviewing our work. Please let us know if these responses address your questions and concerns. If not, we would be more than happy to engage further.

---

> > > ### Author Response · Authors · 2024-11-16
> > > **Response to reviewer M4nr [Part 4/4]**
> > >
> > > ___
> > > ### References:
> > > - Schneider, F., Balles, L., & Hennig, P. DeepOBS: A Deep Learning Optimizer Benchmark Suite. (2019). In International Conference on Learning Representations.
> > > - Dangel, F., Kunstner, F., & Hennig, P. (2020). BackPACK: Packing more into Backprop. In International Conference on Learning Representations.
> > > - Jarrett, D., Yoon, J., Bica, I., Qian, Z., Ercole, A., & van der Schaar, M. (2021). Clairvoyance: A Pipeline Toolkit for Medical Time Series. In International Conference on Learning Representations.
> > > - Jimenez, Carlos E., John Yang, Alexander Wettig, Shunyu Yao, Kexin Pei, Ofir Press, and Karthik R. Narasimhan. (2024). "SWE-bench: Can Language Models Resolve Real-world Github Issues?." In The Twelfth International Conference on Learning Representations.
> > > - Kahl, Kim-Celine, Carsten T. Lüth, Maximilian Zenk, Klaus Maier-Hein, and Paul F. Jaeger. (2024). "ValUES: A Framework for Systematic Validation of Uncertainty Estimation in Semantic Segmentation." In The Twelfth International Conference on Learning Representations.
> > > - Hvarfner, C., Hutter, F., & Nardi, L. (2024). A General Framework for User-Guided Bayesian Optimization. In The Twelfth International Conference on Learning Representations.

---

> > > > ### Author Response · Authors · 2024-11-24
> > > >
> > > > Dear Reviewer,
> > > >
> > > > Thank you again for the time and effort you’ve dedicated to reviewing our submission!
> > > >
> > > > As the rebuttal period is drawing to a close, we wanted to kindly check if our responses have addressed your concerns. If so, we would greatly appreciate it if you could consider updating your score accordingly.
> > > >
> > > > Of course, we remain happy to engage further or clarify any additional questions or comments you may have.
> > > >
> > > > Thank you again for your thoughtful feedback and support during this process!
> > > >
> > > > Best regards,
> > > > Authors of Submission 3712

---

> > > > > ### Comment · Reviewer_M4nr · 2024-11-27
> > > > >
> > > > > Thank you for your detailed response. I appreciate the authors' effort in addressing my concerns, and I have increased my score accordingly. However, I share reviewer ztLg's concern: why not use a package like EconML to run different models independently and select the best one? Therefore, I did not raise my score further.

---

> > > > > > ### Author Response · Authors · 2024-11-27
> > > > > > **Response to Reviewer M4nr**
> > > > > >
> > > > > > Thank you for your thoughtful feedback and for your appreciation of our rebuttal!
> > > > > >
> > > > > > ___
> > > > > >
> > > > > > We agree with the reviewer that a practitioner could, in principle, use a package like EconML to construct different models and select the best one. However, there are (1) many general advantages to using AutoML, and (2) specific motivations for applying it in the context of CATE estimation–we explain both in more detail below. Accordingly, our package is not intended to replace tools like EconML but to provide complementary functionality.
> > > > > >
> > > > > > ### **(1) The importance of automated machine learning in general**
> > > > > >
> > > > > >
> > > > > > First, our work is inspired by the challenges automated machine learning aims to address in general. Automating these processes makes ML algorithms more accessible and user-friendly for practitioners with limited expertise in machine learning, while ensuring adherence to best practices embedded in the automated workflow. Moreover, AutoML can incorporate critical aspects of ML pipelines that are often overlooked, such as feature selection and scaling, which are essential for practical applications.
> > > > > >
> > > > > >
> > > > > > Additionally, formalizing the search process enables the use of more efficient strategies, such as Bayesian optimization, to accelerate the discovery of an optimal pipeline. This process is also guided by robust model validation, ensuring that the resulting ML pipelines are reliable and trustworthy. The entire automated algorithm can be validated across diverse datasets and seamlessly applied to new datasets using the same configuration, ensuring more reliable performance in practice.
> > > > > >
> > > > > >
> > > > > > To draw a parallel from classification, while packages like scikit-learn provide foundational tools, AutoML frameworks such as auto-sklearn build on top of them to automate and enhance the workflow. Similarly, our goal with AutoCATE is to create an automated framework that can build upon tools like EconML, which are designed to address different aspects of the problem.
> > > > > >
> > > > > >
> > > > > > ### **(2) The importance of AutoML for CATE specifically**
> > > > > >
> > > > > >
> > > > > > Automated approaches have been developed for various tasks, including regression, classification, computer vision, and time series forecasting. However, no dedicated solutions currently address the unique challenges of CATE estimation. We argue that automation is particularly crucial for CATE estimation due to the added complexity associated with (1) evaluation and (2) the design of pipelines specifically tailored to CATE estimation.
> > > > > >
> > > > > >
> > > > > > First, simply “selecting the best model” is particularly challenging for CATE estimation. Since the true treatment effects are unknown, we have to rely on evaluation criteria that themselves rely on estimating parameters, necessitating careful optimization and tuning of the underlying models. While existing work, such as Mahajan et al. (2023), provides insights into which criteria perform best and emphasizes the importance of precise model tuning, it does not offer practical procedures for achieving this. In contrast, AutoCATE introduces the first fully automated, end-to-end solution that constructs models for *both* evaluation and estimation.
> > > > > >
> > > > > >
> > > > > > Second, CATE estimation pipelines are significantly more complex than standard supervised learning workflows. Even for experienced data scientists, causal methods introduce additional layers of complexity, limiting their adoption in practice. For instance, a DR-Learner involves up to four distinct ML base-learners: a propensity score model, a control outcome model, a treatment outcome model, and the final effect model. Each of these components can use different ML algorithms, each with its own hyperparameters. This leads to a vast number of potential CATE estimation pipelines to explore. AutoCATE, for example, encompasses 2,187 possible pipelines for CATE estimation *even before considering hyperparameters* (see Appendix B.3). Given this expansive search space, an exhaustive grid search is practically infeasible. Because of this, automated approaches for searching an optimal pipeline are essential in practice.
> > > > > >
> > > > > >
> > > > > > Finally, AutoCATE automates preprocessing steps, such as feature selection and scaling, which leads to improved performance (see Table 10 in the updated paper). While existing CATE estimation packages typically lack such functionality, preprocessing for CATE estimation must also address its unique challenges. For example, a feature that is highly predictive of the outcome, may not be predictive of the treatment effect. By including these steps in our automation, we can guarantee that these preprocessing steps are also finetuned correctly and optimized for the task at hand: accurate CATE estimation.
> > > > > > ___
> > > > > > Once again, thank you for your thoughtful review and for engaging with us throughout this rebuttal! Please let us know if this response fully addresses your concerns or if you have any additional questions.

---

> > > > > > > ### Author Response · Authors · 2024-11-29
> > > > > > >
> > > > > > > Dear Reviewer,
> > > > > > >
> > > > > > > We want to thank you again for your thoughtful feedback and for already updating your score—it truly means a lot to us!
> > > > > > >
> > > > > > > We wanted to kindly follow up to ask if our latest response addressed your remaining concerns. If so, we would greatly appreciate it if you might consider updating your score further. If there are any unresolved questions or areas where we can further clarify, we would be more than happy to do so—please do not hesitate to let us know!
> > > > > > >
> > > > > > > Thank you again for your time, engagement, and invaluable feedback. Wishing you a very Happy Thanksgiving!
> > > > > > >
> > > > > > > Best regards,
> > > > > > > The Authors of Submission 3712

---

> ### Author Response · Authors · 2024-12-02
>
> Dear Reviewer,
>
> As the rebuttal period is now coming to an end, we wanted to send a final reminder to kindly ask if our responses have addressed your concerns. If so, we would be deeply grateful if you could again consider updating your score.
>
> Thank you again for your thoughtful feedback and engagement throughout this process!
>
> Best regards,
>
> The Authors of Submission 3712

---

### Official Review · Reviewer_usPL · 2024-11-04

**Soundness:** 4
**Presentation:** 4
**Contribution:** 4
**Rating:** 8
**Confidence:** 3

**Summary:**

This paper introduces a software development framework: AutoCATE, which aims at automating the estimation of Conditional Average Treatment Effects (CATE) for real-world applications. It addresses the unique challenges in CATE estimation: the absence of ground truth for counterfactuals and the complexity of causal ML pipelines, by formalizing the pipeline search as a counterfactual Combined Algorithm Selection and Hyperparameter (CASH) optimization problem.

**Strengths:**

**Originality**: The paper proposes an innovative approach to address the specific complexities of CATE estimation with an automated solution tailored for non-expert practitioners. By implementing a counterfactual CASH optimization, it well addresses the gaps in the CATE literature.

**Quality**: The paper is well-structured, from its technical details to empirical evaluations. The effects of the proposed AutoCATE have been well-demonstrated through extensive comparisons with existing CATE estimation methods, supporting the validity of their claims.

**Clarity**: The overall writing of the paper is clear and easy to follow. Regarding its technical details, the problem formulation and the proposed three-stage solution are well-explained, making complex concepts easy to understand. The figures (e.g., pipeline flow and risk comparisons) and tables add clarity, effectively supporting the text.

**Significance**: AutoCATE is highly relevant for domains requiring accurate causal inference, as it democratizes access to CATE estimation methods by reducing the technical expertise needed. This can drive real-world adoption of CATE estimation in fields like healthcare, where accurate treatment effect estimation has significant implications.

**Weaknesses:**

**Computation Complexity**: This is my major concern. How long would it take for AutoCATE to find an optimal ML pipeline given a dataset (e.g. the smallest and the biggest datasets you used in your experiments)? How would the size of the dataset impact the time of finding the optimal ML pipeline?

**Algorithm Coverage**: From the demonstration of the framework, only basic ML algorithms are covered in your software. I wonder are there more sophisticated ML algorithms that can be used for CATE estimation? If yes and if these algorithms are integrated into the entire framework, how will it impact the computation complexity?

**Questions:**

Please refer to weakness.

---

> ### Author Response · Authors · 2024-11-16
> **Response to reviewer usPL [Part 1/2]**
>
> Thank you for your thoughtful and detailed feedback on our work! Below, we address each point raised in your review.
> ___
>
> ## Computational Complexity
>
> We agree that computational efficiency is an important consideration for real-world adoption. The time required to run AutoCATE depends on several factors, many of which can be configured to balance computation time and performance:
> - **Metalearners:** the complexity of the metalearners (e.g. DR-Learner is slower than S-Learner)
> - **Baselearners:** the complexity of the baselearners (e.g. random forest is slower than linear regression)
> - **Number of trials:** the number of trials to optimize the models for evaluation
> - **Data set**: More data or a higher dimensionality would naturally increase computation time.
>
> We have aimed to make AutoCATE as efficient as possible using efficient ML algorithms in scikit-learn and parallelization in optuna. To give a concrete sense of AutoCATE’s runtime, we include the following computation times for different data sets below and in the updated paper. Although some time is required, running our framework locally is certainly feasible for small to moderate data sets. These experiments were conducted locally, on a machine with an AMD Ryzen 7 PRO 4750U processor (1.70 GHz), 32 GB of RAM, and a 64-bit operating system.
>
> ### **Time required to run AutoCATE**
> #### (BestBase-BestMeta configuration with 50 trials for evaluation and 50 trials for estimation)
> |                              |   **IHDP**  |    **ACIC**   |    **Twins**   |     **News**     |
> |-----------------------------:|:-----------:|:-------------:|:--------------:|:----------------:|
> |                              | n=747; d=25 | n=4,802; d=58 | n=11,984; d=46 | n=5,000; d=3,477 |
> | _Average time required_ | 1'21"       | 6'00"         | 29'38"         | 6'49"            |
>
> - **Action taken:** We have included a discussion of these results in our work.
>
> We acknowledge that further efforts to improve computational efficiency would benefit the applicability of our framework, as noted in the “Future Work” section. For example, we could use insights from AutoML for other applications, such as those described in Wang et al. (2021), to come up with new strategies for faster optimization. Tailored approaches for CATE estimation could also be developed. Figure 10 compares the time complexity and accuracy of different metalearners, illustrating how search space configuration can offer a useful tool for controlling this trade-off. In practice, visualization tools provided in AutoCATE, based on optuna, can help judge whether additional trials will have an effect.
>
> We hope that our work and the associated software package motivates the community to contribute toward more time-efficient solutions.
>
>
> ## Algorithm Coverage
>
> The methods included in AutoCATE were chosen for their popularity, versatility, and efficiency. Nevertheless, we agree with the reviewer that including more advanced ML methodologies tailored for CATE estimation would enhance our framework.
>
> In future versions of the software package, we plan to extend the included methods and provide an API to allow users to seamlessly integrate custom methodologies into the AutoCATE search space. This would enable researchers and practitioners to adapt AutoCATE for their specific use cases with minimal effort.
>
> Tailored neural approaches are particularly promising approaches within this context (see e.g., Shalit, Johansson & Sontag, 2017; Yoon, Jordon & van der Schaar, 2018; Shi, Blei & Veitch, 2019). Automated approaches would be especially useful here to help optimize the design of these neural networks. For example, by tuning the number of layers, by deciding to add balanced representations or not, or by optimizing how the treatment variable is added. We explicitly identify neural architecture search as a key area for future research in our paper.
>
> We also recognize that advanced algorithms may increase AutoCATE’s computational complexity. However, we believe AutoCATE presents an important opportunity for the community: to evaluate algorithms not only on the precision of their estimates but also on their computational efficiency, an aspect often overlooked in existing work.
>
> ___
>
> Again, we greatly appreciate your thoughtful feedback and time in reviewing our work! Please let us know if there are any remaining concerns.

---

> > ### Author Response · Authors · 2024-11-16
> > **Response to reviewer usPL [Part 2/2]**
> >
> > ### References
> > - Wang, C., Wu, Q., Weimer, M., & Zhu, E. (2021). FLAML: A fast and lightweight automl library. Proceedings of Machine Learning and Systems, 3, 434-447.
> > - Shalit, U., Johansson, F. D., & Sontag, D. (2017, July). Estimating individual treatment effect: generalization bounds and algorithms. In International conference on machine learning (pp. 3076-3085). PMLR.
> > - Yoon, J., Jordon, J., & Van Der Schaar, M. (2018, February). GANITE: Estimation of individualized treatment effects using generative adversarial nets. In International conference on learning representations.
> > - Shi, C., Blei, D., & Veitch, V. (2019). Adapting neural networks for the estimation of treatment effects. Advances in neural information processing systems, 32.

---

> > > ### Author Response · Authors · 2024-11-24
> > >
> > > Dear Reviewer,
> > >
> > > Thank you again for the time and effort you’ve dedicated to reviewing our submission!
> > >
> > > As the rebuttal period is drawing to a close, we wanted to kindly check if our responses have addressed your concerns. If so, we would greatly appreciate it if you could consider updating your score accordingly.
> > >
> > > Of course, we remain happy to engage further or clarify any additional questions or comments you may have.
> > >
> > > Thank you again for your thoughtful feedback and support during this process!
> > >
> > > Best regards,
> > > Authors of Submission 3712

---

> ### Author Response · Authors · 2024-11-27
>
> Dear Reviewer,
>
> As the rebuttal period is nearing its end, we wanted to kindly follow up to see if our responses have addressed your concerns. If they have, we would be grateful if you could consider updating your score accordingly.
>
> If there are any remaining questions or points you would like us to address, please do not hesitate to let us know. We are more than happy to provide further clarifications or engage in additional discussion.
>
> Thank you once again for your time and thoughtful feedback throughout this process!
>
> Best regards,
>
> Authors of Submission 3712

---

> > ### Author Response · Authors · 2024-11-29
> >
> > We hope this message finds you well! As the rebuttal period has been extended, we wanted to follow up once more to kindly ask if our responses have addressed your concerns. If they have, we would be truly grateful if you could consider increasing your score, as two other reviewers have already done after the rebuttal.
> >
> > Of course, if there are any outstanding points you would like us to address further, please do not hesitate to let us know—we are happy to provide additional details.
> >
> > Thank you once again for your thoughtful engagement and for the time devoted to reviewing our work. Wishing you a very happy Thanksgiving!
> >
> > Best regards,
> >
> > The Authors of Submission 3712

---

> > > ### Comment · Reviewer_usPL · 2024-12-02
> > > **Thank you for your response**
> > >
> > > My concerns are addressed, although I'm not an expert from the field, I'm willing to increase my score.

---

> > > > ### Author Response · Authors · 2024-12-02
> > > >
> > > > Dear Reviewer,
> > > >
> > > > Thank you very much for your thoughtful feedback and for engaging with us during the rebuttal process. We are glad our responses addressed your concerns and deeply appreciate you increasing your score.
> > > >
> > > > We will ensure that your input is reflected in the final version. Thank you again for helping improve our work!
> > > >
> > > > Best regards,
> > > >
> > > > The Authors of Submission 3712

---

### Official Review · Reviewer_riY4 · 2024-11-04

**Soundness:** 2
**Presentation:** 3
**Contribution:** 3
**Rating:** 8
**Confidence:** 4

**Summary:**

The authors propose a pipeline for automating the several design choices required for CATE estimation; from preprocessing datasets to different risk measures for model selection. The pipeline is divided into three stages corresponding to the following three questions; what risk measure should be used for model selection, what CATE estimators should be trained, and finally how should be select over the trained CATE estimators and combine them for better generalization. The authors conduct experiments on widely used benchmarks and present interesting insights regarding the numerous design choices in CATE estimation.

**Strengths:**

- The authors have done a really good job at covering nearly all the design choices involved in CATE estimation. Specifically, analyzing the role of preprocessing datasets and dataset splits for training/evaluation has not been done in prior works. Further, the authors experiment with novel strategies for model selection with multiple risk measures and ensembling of CATE estimators.

- The scale of the empirical study is quite comprehensive; experiments involve a variety of meta-learners, base-learners, and risk measures. This makes their findings interesting and significant for practitioners and future work, and their software package should also make it easy for practitioners to adopt the proposed pipeline.

- The paper overall is well written and organized which makes it easy to follow and understand main results. The experiment results are clearly presented with good discussion around them. I especially like their comparisons with the findings from prior benchmarking studies for CATE model selection.

**Weaknesses:**

- I have concerns regarding the lack details for some key aspects of their empirical study. For experiment regarding risk measures in Section 5.2, how were the nuisance models associated with the risk measures selected? Also, what is the underlying set of trained CATE estimators over which authors are performing model selection via different risk measure? The caption inTable 1 state "Results for 50 evaluation trials and 50 estimation trials with a T -Learner and gradient boosting". Does this imply that model selection was done over only T Learners? That would be a serious issue as we want to select over a diverse set of CATE estimators to benchmark the different risk measures. For example, due to the congeniality bias (Curth & van der Schaar (2023), if the CATE estimators only involve T Learners, then T risk should do better than other risk measures. I would like the authors to clarify this point and provide clear details regarding this in the paper. The same issue is repeated with ensembling experiment in section 5.4, as the caption in Table 3 suggests the CATE estimators trained were only T Learners.

- Regarding experiment in section 5.5, the authors should follow the procedure of AutoML to tune S/T Learner (Mahajan et al. 20023) instead of manual grid search. This would ensure stronger baselines and a fair comparison with them. Similarly, the authors can construct meta-learners with nuisance model trained via AutoML (Mahajan et al. 20023), and that could serve an alternative set of CATE estimators for experiment in section 5.3 (Estimation) as well. For example, the BestBase estimator currently involves a manual search over a grid of different algorithms and hyperparameters, but this could be automated via AutoML.

- I am not sure what are the main conclusions from the experiments with combined risk measures? The authors did not experiment with many combinations, and only considered combining T & DR risk and different T risk. So the experiments are not exhaustive which makes it hard to interpret what the main trend should be and what recommendations can be made. Similar comment for ensembling with multiple risk measures; I think the strategy of combining risk measures is the most novel aspect of the work, so analyzing it in depth would make the paper strong.

- The experiments with ensembling in section 5.4 are good but they are missing the ensembling strategy of using softmax based on risk measures from Mahajan et al. 2023. I would recommend to have some discussion around the potential advantages/disadvantages of their proposed ensembling strategy over the softmax risk measure based ensembling. I understand experimenting with every possible ensembling strategy is not feasible by the rebuttal period, but I would encourage the authors to include it in their package for later.

- Finally, I find the counterfactual Combined Algorithm Selection and Hyperparameter optimization (CASH) formulation a bit misleading. The problem studied by the authors is essentially model selection for CATE estimation, as stated in prior works (Schuler et al. 2018,  Curth & van der Schaar (2023), Mahajan et al. 2023). I am not sure why the authors want to reformulate this? It makes the connection with the prior works weaker as it seems to give an impression that the authors are solving a different (and novel) problem.

**Questions:**

- How are the multiple risk measures combined? Do we take the average of risk measures, like average of T and DR risk in the experiments?

- How do the authors obtain the best meta-learner or best base-learner? Is it based on how well they fit the observational data?

Minor comments

- In lines 109-110 authors state that prior works haven't explored AutoML for CATE estimation. This should be changed to account for the work by Mahajan et al. 2023, as acknowledged by the authors themselves ahead in line 144

- In line 258, the authors state that there are no prior works for ensembling CATE estimators, but they have acknowledged the work by Mahajan et al. in line 123 for ensemble selection. Please update the text to make it more consistent.

- It will be good to add equations to explain the proposed ensembling strategy in section 4.3

- It would be nice to have statistics regarding the scale of the empirical study before section 5; like how many risk measures, how many meta-learners and base-learners for estimation, how many estimators are included for the model selection study, etc.

---

> ### Author Response · Authors · 2024-11-16
> **Response to reviewer riY4 [Part 1/3]**
>
> Thank you for your detailed review and constructive feedback! Below, we address your concerns point by point, structured into the following sections for clarity: experimental setup, automatically finding the best model and benchmarking, softmax ensembling, CASH formulation, combining risk measures, and minor comments.
>
> ___
>
> ## Experimental setup
>
> First, we would like to clarify the captions in Tables 1 and 3: _“Results for 50 evaluation trials and 50 estimation trials with a T-Learner and gradient boosting”_.
> For these results, AutoCATE uses 50 AutoML trials to optimize the ML pipelines underlying the risk measures, to be used for **evaluation**. More specifically, a random search was performed over 50 ML pipelines, in this case containing different gradient boosting models, for each estimate required in the risk measure.
> Similarly, we use 50 AutoML trials to optimize the ML pipeline for CATE **estimation**. In the case of Tables 1 and 3, we indeed only searched over T-Learners built using gradient boosting models, as we wanted to limit the complexity and the size of our search space.
>
> - **Action taken:** We appreciate that this may not have been sufficiently clear. In the updated paper, we have included a more detailed explanation of the captions in Section 5.1.
>
> We appreciate the reviewer’s concern regarding congeniality bias. To analyze and mitigate this, we have conducted additional experiments (we refer to Tables 7 (a) and (b) in the updated paper for more details).
> - **S-Learners only**: To reduce the potential impact of congeniality bias, we compare different model selection criteria when AutoCATE is constrained to only using S-Learners. As the S-risk is not considered, congeniality bias should not be an issue here.
> ### Comparing model selection for S-Learners only
>
> |         | **DR** | **F** | **IF** | **kNN** | **R** | **T** | **U** | **Z** |
> |--------:|:------:|:-----:|:------:|:-------:|:-----:|:-----:|:-----:|:-----:|
> |  _IHDP_ |  3.21  |  3.64 |  4.60  |   3.11  |  3.48 | **3.10** |  3.62 |  4.12 |
> |  _ACIC_ |  1.61  |  1.79 |  2.07  |   1.88  |  1.73 | **1.58** |  1.85 |  2.16 |
> | _Twins_ |  .328  |  .328 |  .347  |   .320  |  .325 | **.320** |  .321 |  .330 |
> |  _News_ |  2.47  |  2.51 |  2.97  |   2.94  |  2.76 |   2.46   |  2.78 |  2.99 |
>
> - **BestMeta configuration**: We additionally included experiments where AutoCATE searches over all the best metalearners (BestMeta). Congeniality bias would actually complicate results for some learners in this setting: for example, the T-risk may prefer a T-Learner even when it is not the optimal model.
> ### Comparing model selection for selected metalearners (BestMeta)
> |         | **DR** | **F** | **IF** | **kNN** | **R** | **T** | **U** | **Z** |
> |--------:|:------:|:-----:|:------:|:-------:|:-----:|:-----:|:-----:|:-----:|
> |  _IHDP_ |  **2.07**  |  3.43 |  5.75  |   2.11  |  3.45 |  2.17 |  3.18 |  4.38 |
> |  _ACIC_ |  1.40  |  1.87 |  2.24  |   1.97  |  1.57 |  **1.35** |  1.79 |  2.16 |
> | _Twins_ |  .328  |  .327 |  .384  |   **.324**  |  .328 |  .326 |  .344 |  .348 |
> |  _News_ |  **2.42**  |  2.60 |  2.95  |   2.42  |  2.75 |  2.43 |  2.78 |  2.77 |
>
> These findings are in line with our original conclusions: the $DR$, $kNN$, and $T$-risks seem to work relatively well across settings and datasets. Although congeniality bias may be present, we do not find that it has a strong impact on the $T$-risk’s performance. Compared to Curth & van der Schaar (2023), we more carefully tune the ML pipelines underlying risk measures. We hypothesize that this could help to mitigate congeniality bias.
>
> - **Action taken:** We have included these additional experiments in our paper, discussing their results in the context of congeniality bias.

---

> > ### Author Response · Authors · 2024-11-16
> > **Response to reviewer riY4 [Part 2/3]**
> >
> > ## Automatically finding the best model and benchmarking
> >
> > The reviewer asks: _“How do the authors obtain the best meta-learner or best base-learner?”_ We want to stress that AutoCATE builds the best ML pipeline completely in a manner that is completely **automated** and **end-to-end**. First, AutoCATE optimizes different ML pipelines to construct an accurate risk measure for model selection. Then, ML pipelines are built for CATE estimation, searching over different meta- and baselearners, in order to automatically optimize the risk measure from the first step. In this way, AutoCATE offers the first automated, end-to-end procedure that integrates evaluation, estimation, and ensembling in a unified workflow–all wrapped in a single “fit” procedure (see Section 4.5).
> >
> > AutoCATE adopts an autoML approach for tuning ML pipelines in the first two stages: evaluation (constructing the risk measures) and estimation. In contrast, Mahajan et al. (2023) focus on the evaluation phase (i.e., model selection), using AutoML to tune risk measures but not the final CATE estimators. In that sense, procedures such as those in Mahajan et al. (2023)’s are naturally included within our broader framework and can be viewed as ablations of our more comprehensive configuration. AutoCATE includes offers a fully automated solution across all stages and offers larger search spaces–with more meta- and baselearners– and additional design choices.
> >
> > As part of our framework, the “BestBase” configuration limits the search space to a subset fo baselearners which are known to perform well for tabular data in general, with the goal of enhancing the search’s efficiency and performance. Importantly, only the search space is constrained in this way, the search itself remains completely automated.
> >
> > ## Softmax ensembling
> >
> > We agree that this represents a useful alternative strategy and have included it in our framework. Thank you for the suggestion!
> >
> > -  **Action taken :** We have included empirical results comparing our stacking procedure (COP, based on a constrained optimization problem) and theirs (Softmax). These results have been included in Table 3 of the updated paper.
> >
> > For your convenience, we include results for a single T-Risk below:
> > ### Comparing approaches for stacking CATE estimators with a single T-risk objective
> > |         | **COP**         | **Softmax**    |
> > |--------:|:---------------:|:--------------:|
> > | _IHDP_  | 1.96 $\pm$ .34  | 2.83 $\pm$ .51 |
> > | _ACIC_  | 1.42 $\pm$ .09  | 1.33 $\pm$ .09 |
> > | _Twins_ | .344 $\pm$ .00  | .331 $\pm$ .00 |
> > | _News_  | 2.33 $\pm$ .06 | 2.33 $\pm$ .06 |
> >
> > ## CASH formulation
> >
> > We framed our work as addressing a counterfactual Combined Algorithm Selection and Hyperparameter optimization (CASH) problem to emphasize its nature as a _search problem_ that extends _beyond_ the problem of model selection as considered in prior work. AutoCATE performs automated searches across a range of possible ML pipelines for constructing the risk measures and CATE estimators. The outcome of this search is an optimized pipeline or ensemble for CATE estimation.
> >
> > In other words, prior work focuses only on finding the best criterion for model selection. AutoCATE’s approach unifies these stages within an end-to-end, automated framework designed to find the best possible ML pipeline for CATE estimation. By adopting the term CASH, we align with established AutoML terminology for describing this type of search problem, reinforcing the connection between our approach and broader research on AutoML.
> >
> > That said, if the reviewer still finds this terminology unclear or misleading, we are happy to consider alternative phrasing to improve clarity and consistency with existing literature.
> >
> > ## Combining risk measures
> > We first describe how different risk measures are combined and then discuss our results.
> >
> > ### (1) _”How are the multiple risk measures combined?”_
> >
> > We consider different strategies for combining risk measures in our work. As the reviewer points out, we could indeed take the average of the (normalized) risk measures. Additionally, we also consider different strategies, such as averaging the risk measure rankings or their distance to the origin. The rankings would be more robust to risk measure outliers. Finally, we also include the possibility to select all Pareto optimal points and the stacking procedure(s). These strategies are compared empirically in Table 3.
> >
> > -  **Action taken**: We appreciate that these strategies may not have been explained in sufficient detail. We have added a more detailed explanation for each, including equations, in the appendix.
> >
> > Related to this, Table 1 (b) indeed uses the average of the (normalized) risk measures.
> >
> > - **Action taken**: We have updated the description of the captions to more clearly reflect this.

---

> > > ### Author Response · Authors · 2024-11-16
> > > **Response to reviewer riY4 [Part 3/3]**
> > >
> > > ### (2) Results and conclusions
> > >
> > > Our analysis of combining risk measures aims to provide an initial exploration and validation of this approach. Specifically, we tested combinations of well-performing individual risk measures–T and DR, as well as T, DR and kNN (see Table 1a)--and considered multiple versions of the T-risk measure. Our hypothesis was that depending on different risk measures would be more robust than using a single one. The primary conclusion from these experiments is that this approach shows promise, though no strategy consistently outperforms using a single T-Risk.
> > >
> > > As the reviewer notes, many more combinations are possible. For example, with 8 risk measures, there are 28 pairwise combinations, 56 combinations of three, and so on. Additionally, more advanced strategies could be explored, such as leveraging risk measure correlations (see Figure 14a) and building more advanced ensembles (e.g. using stacking). Given our success with ensembling strategies for CATE estimation, we believe these directions are promising and could lead to more robust performance.
> > >
> > > If the reviewer has any particular combinations of risk measure of interest, we would be more than happy to include them in our experiments. Nevertheless, we consider an exhaustive exploration of all possible combinations to be outside the scope of this work. Instead, our goal is to highlight the potential of this approach and provide a foundation for future research. We hope that our findings and the AutoCATE software package will encourage the community to further explore these ideas.
> > >
> > > ## Minor comments
> > > Thank you for pointing these out! We have made the following changes:
> > > - Mahajan et al. (2023) use AutoML for CATE model _selection_, but not for model _estimation_. We have made this distinction more clear in the revised version.
> > > - We have updated line 258 to point out Mahajan et al. (2023)’s work on ensembling.
> > > - We have included an additional section in the appendix of the updated paper, explaining all ensembling strategies with equations.
> > > - We mention that we run experiments across 247 distinct data sets. Thank you for this suggestion! Additionally, we would like to point out that the building blocks supported in AutoCATE are shown in Figure 2, and we describe the size of the estimation search space in Appendix B.3. We would be more than happy to include this information in some other manner if the reviewer believes this would strengthen our work.
> > >
> > > ___
> > >
> > > Thank you again for your time and effort in reviewing our work! We hope that these updates adequately address your concerns. If there are any remaining points, we would be more than happy to address them.

---

> > > > ### Author Response · Authors · 2024-11-24
> > > >
> > > > Dear Reviewer,
> > > >
> > > > Thank you again for the time and effort you’ve dedicated to reviewing our submission!
> > > >
> > > > As the rebuttal period is drawing to a close, we wanted to kindly check if our responses have addressed your concerns. If so, we would greatly appreciate it if you could consider updating your score accordingly.
> > > >
> > > > Of course, we remain happy to engage further or clarify any additional questions or comments you may have.
> > > >
> > > > Thank you again for your thoughtful feedback and support during this process!
> > > >
> > > > Best regards,
> > > > Authors of Submission 3712

---

> > > > > ### Comment · Reviewer_riY4 · 2024-11-25
> > > > >
> > > > > Thanks a lot for your detailed response! Thanks for the additional experiments, I appreciate the inclusion of the softmax ensembling approach and experiments regarding congeniality bias! I am also satisfied with the response on combining risk measures, indeed I don't expect authors to run all possible combinations. While some more interesting trend based on combining risk measures might have been nice, but I agree it is like a first step to promote future research on this. Also, it definitely adds to the flexibility of the package, users can experiment with combining different risk measures. Hence, I have adjusted my rating to 6 for now, and stating my unresolved concerns ahead.
> > > > >
> > > > > Regarding the experiment setup for step 1 of finding a good risk measure, why don't the authors experiment with a more diverse collection of base estimators and meta-learners? I understand that the point of the paper unlike prior works is not to shed novel insights on model selection metrics, but I think the setup can be improved for future work post rebuttal. I would encourage authors to look at the setup in the prior works (Curth & van der Schaar (2023), Mahajan et al. 2023) for the underlying list of CATE estimators trained for model selection.
> > > > >
> > > > > Which bring me to my second question, which is still unresolved: Automatically finding the best model and benchmarking. In step 2, authors basically conduct a grid search over base learners and meta learners and determine the best model using a risk measure they inferred from Step 1. It is not entirely correct to say this is the first approach to do so, as one could use AutoML (FLAML: https://microsoft.github.io/FLAML/) to search over different nuisance model and construct CATE estimators. This involves essentially fixing the risk measure as mu score (how good nuisance models fit the data), but instead of a manual grid search, use AutoML to infer the best model class and associated hyperparameters. My understanding is that this approach is not included in the author's framework. Also, it is incorrect to say that Mahajan et al. 2023 only used AutoML for model selection, rather they used FLAML to optimize the nuisance models. Could the authors clarify about this technique of finding optimal CATE estimators without explicit grid search? This is especially important for the benchmarking AutoCATE experiment, as the S/T learner can be tuned with mu-risk without a grid search, but using FLAML. Please refer to the  second point in the weakness section of my original review, and I am happy to clarify it more if needed.
> > > > >
> > > > > This point above makes me still question the CASH formulation; prior works on model selection were also creating pipelines for automatic selection of CATE estimators! If the only additional step after inferring a good risk measure is to run a grid search on different base-learners and meta-learners, then I don't really see what is the contribution there? Is that contribution really significant enough to allow for a new formulation? There can be more smarter ways of finding the optimal CATE estimators than grid search,  as stated above on the use of FLAML for inferring both the best model class and hyperparameters for base-learners.

---

> > > > > > ### Author Response · Authors · 2024-11-26
> > > > > > **Response to reviewer riY4 [Part 1/3]**
> > > > > >
> > > > > > Thank you for your thoughtful feedback and positive response to our updates!
> > > > > > We now have a clearer understanding of your initial concerns based on your reply and have provided more detailed responses to each point below.
> > > > > > ___
> > > > > >
> > > > > > ## **Collection of base- and meta-learners**
> > > > > >
> > > > > > In Section 5.2, we limit the scope of base- and meta-learners to simplify our experimental setup and isolate the *effect of model evaluation*. This approach allows us to focus on key questions related to this stage, such as: (1) which evaluation criteria are most effective, (2) how much data should be allocated for evaluation versus estimation, and (3) how different metrics impact the results.
> > > > > >
> > > > > > Building on these insights, we expand our experiments to include more diverse base- and meta-learners in Sections 5.3 and 5.4. Our search spaces there encompass the most commonly used ML base-learners and, to the best of our knowledge, all established meta-learners. In this regard, our collections are more comprehensive than those in prior works, such as Curth & van der Schaar (2023) and Mahajan et al. (2023).
> > > > > >
> > > > > > AutoCATE encompasses a wide range of design choices for identifying an optimal pipeline for CATE estimation. To study these choices systematically, we organized them into three stages and analyzed each stage independently. To do this effectively, we initially reduced the complexity of our setup by keeping other design choices as simple as possible. Because of this, we limited the baselearners considered for comparing the model selection criteria. That said, if the reviewer believes we should incorporate more base- and meta-learners in Section 5.2, we are happy to include additional experiments to test specific hypotheses.
> > > > > >
> > > > > > ## **Automatically finding the best model and benchmarking**
> > > > > >
> > > > > > We address your second question in three parts. First, we explain how AutoCATE automatically identifies the best model. Second, we elaborate on the differences between our work and that of Mahajan et al. (2023). Third, we provide details on the benchmarking process within this context.
> > > > > >
> > > > > > ### **How to automatically obtain the best pipeline?**
> > > > > >
> > > > > > First, let us clarify how we identify the best pipeline. Instead of performing a *complete grid search* across all possible combinations of base- and meta-learners, we employ a *random search*. This distinction is crucial, as our work focuses on finding a good CATE estimator in a way that is feasible in practice. A full grid search is rarely practical due to the enormous number of potential pipelines—AutoCATE includes 2,187 possible pipelines for CATE estimation **even before considering hyperparameter variations**. By contrast, the random search allows us to explore how to efficiently identify a strong CATE estimator **in practical scenarios**. For instance, we investigate how many random search trials are necessary to achieve good performance in practice (e.g., see Figures 3, 5, and 7). In these figures, the number of trials corresponds to how many of these random pipelines are trained.
> > > > > >
> > > > > > The above procedure, based on random search, serves as a (rudimentary) AutoML approach. While more sophisticated methods (e.g., those used in FLAML) are certainly possible, we intentionally kept the setup simple to focus on the design choices **unique to CATE estimation**. That said, AutoCATE is highly flexible and supports more advanced search strategies, as it is fully compatible with all optimization methods available in Optuna (Akiba et al., 2019).

---

> > > > > > > ### Author Response · Authors · 2024-11-26
> > > > > > > **Response to reviewer riY4 [Part 2/3]**
> > > > > > >
> > > > > > > ### **The difference with respect to Mahajan et al. (2023)**
> > > > > > >
> > > > > > > Second, let us clarify the differences between Mahajan et al. (2023) and our work. While both use AutoML, the approaches differ significantly in *how* AutoML is applied and *why* it is used.
> > > > > > >
> > > > > > > Mahajan et al. (2023) aim to gain insight into **model selection criteria for CATE estimation**. To compare these criteria, they evaluate which criterion is most effective at selecting a CATE estimator from a *large pool of candidate estimators*. To build this pool of 415 well-performing candidates, they use AutoML to optimize the nuisance parameters underlying these models. However, for most meta-learners, the final CATE estimators themselves are not optimized with AutoML in their work. As these are not standard supervised models, the mu-risk criterion is not applicable and standard AutoML approaches like FLAML cannot be directly employed. Instead, they construct various CATE estimators by combining the optimized nuisance parameters to populate their pool of candidates.
> > > > > > >
> > > > > > > While this approach is valuable for comparing model selection criteria, several questions remain regarding the practical process of finding a CATE estimator. How should we approach tuning the final model? How can we do so efficiently (e.g., how long should nuisance parameters be optimized)? Moreover, the nuisance parameters optimized with AutoML may not always serve as the best building blocks for final CATE estimators. For instance, very small or very large propensity scores can lead to significant instabilities during training for some meta-learners.
> > > > > > >
> > > > > > > These unresolved questions motivate our work, which has a different goal: we investigate **how we can efficiently find a good CATE estimator**. While an exhaustive grid search with extensive tuning of each submodel, like the approach in Mahajan et al. (2023), may be *effective* in practice, it is certainly not *efficient*. In contrast, we explore a range of design choices related to the various search problems involved in automatically finding a CATE estimator. Furthermore, our approach uses an automated approach to directly identify the *final CATE estimator*, rather than submodels that underlie it. Finally, we also address aspects of the ML pipeline that were overlooked by Mahajan et al. (2023), such as preprocessing and probability calibration. Automating these steps is crucial in practice, as demonstrated in Table 10 of the updated paper.
> > > > > > >
> > > > > > > Related to this, the reviewer correctly points out that Mahajan et al. (2023) use AutoML to tune nuisance models for CATE estimators. Our earlier statement that this is limited to model selection only was indeed not correct. We will update our manuscript accordingly. Our apologies for the confusion and thank you for pointing this out!
> > > > > > >
> > > > > > > ### **Benchmarking**
> > > > > > >
> > > > > > > Third, let us explain our approach for benchmarking AutoCATE. Essentially, AutoCATE automatically searches across different base- and meta-learners based on a T-risk. An alternative, equally viable approach–which serves as our benchmark–would involve using a mu-risk. However, this strategy can only be applied to S- and T-Learners. Therefore, we compare AutoCATE against various S- and T-Learners, each tuned based on the mu-risk.
> > > > > > >
> > > > > > > For both AutoCATE and the benchmarks, we use a random search to focus on the effect of the search spaces involved and evaluation criteria considered. More sophisticated search strategies (e.g., those used in FLAML) could be employed instead of random search, but we chose to use random search for all strategies to facilitate a straightforward comparison. Given that AutoCATE has the largest search space, we believe it would particularly benefit from more advanced search algorithms. As mentioned earlier, AutoCATE supports a range of sophisticated search strategies available in Optuna.
> > > > > > >
> > > > > > > Finally, a more advanced benchmark would involve searching across S- and T-Learners, as well as different base-learners, using the mu-risk. This approach, which we plan to include in a future version of AutoCATE, is more complex and not currently supported by even advanced AutoML libraries like FLAML, which only include S-Learners. This highlights the need for tailored approaches to CATE estimation.

---

> > > > > > > > ### Author Response · Authors · 2024-11-26
> > > > > > > > **Response to reviewer riY4 [Part 3/3]**
> > > > > > > >
> > > > > > > > ## **CASH formulation**
> > > > > > > >
> > > > > > > > We use a different formulation to highlight the distinct goal of our work compared to prior studies, such as Mahajan et al. (2023), as discussed above. A key motivation of our work is related to practical feasibility: how can we efficiently find a good 6CATE estimator in practice? The fact that Mahajan et al. do not address important questions (as noted earlier) underscores the importance of this practical perspective. The approach we advocate aligns with the perspective commonly used in AutoML. Specifically, while the CASH formulation is a novel term in the context of CATE estimation, it is a standard concept for this type of problem in AutoML, which is why we chose to adopt it. In conclusion, while we acknowledge that our problem is related to prior work on CATE estimation, we believe that a precise problem formulation is crucial for clearly stating the goals and contributions of our work.
> > > > > > > >
> > > > > > > >
> > > > > > > >
> > > > > > > >
> > > > > > > > ___
> > > > > > > > Thank you once again for your active engagement in this rebuttal–this is very much appreciated! Please let us know if these responses address your concerns. If not, we would be happy to discuss them further.
> > > > > > > >
> > > > > > > >
> > > > > > > > ___
> > > > > > > >
> > > > > > > > ### References
> > > > > > > >
> > > > > > > > - Akiba, T., Sano, S., Yanase, T., Ohta, T., & Koyama, M. (2019, July). Optuna: A next-generation hyperparameter optimization framework. In Proceedings of the 25th ACM SIGKDD international conference on knowledge discovery & data mining (pp. 2623-2631).

---

> ### Comment · Reviewer_riY4 · 2024-11-27
>
> Thanks a lot for engaging in discussion with me! I would request the authors to include these details regarding the comparison with prior works and specifics about how they conducted grid search, other alternatives like Flaml that could be used, etc. in their paper. It is important as it provides full context to the reader, so please add most of the points/clarification from our discussions in the paper.
>
> I am happy with the responses and the engagement during rebuttal! I believe this work would make good impact in the community; I have increased my score and vouch for acceptance.

---

> > ### Author Response · Authors · 2024-11-29
> >
> > Thank you so much for your thoughtful feedback and active involvement throughout the rebuttal process! We sincerely appreciate the time and effort you dedicated to engaging with us, and for your continued constructive suggestions on improving our paper.
> >
> > We are delighted to hear that you found our responses helpful and that you believe our work will make a meaningful impact on the community. We will certainly include your suggestions and clarifications from our discussion in the revised version.
> >
> > Your insights and engagement have contributed a lot to improving the quality and clarity of our work, and we are deeply grateful for your support. Thank you once again for your encouraging feedback and for increasing your score—we truly appreciate your confidence in our work!
> >
> > The authors

---

### Official Review · Reviewer_mUTg · 2024-11-07

**Soundness:** 3
**Presentation:** 3
**Contribution:** 1
**Rating:** 3
**Confidence:** 4

**Summary:**

The paper proposes an end-to-end pipeline for CATE estimations and conducts a comprehensive comparison with existing methods.

**Strengths:**

I think the paper has comprehensive experiments and reproducible code.

**Weaknesses:**

Novelty and significance of the paper:
The methods in each stage of the autoCATE pipeline are not new and the pipeline itself is not novel either. The paper does conduct a comprehensive experiments, but again most of the similar comparisons are already covered in the existing papers. The author claims the stacking is novel, but from my knowledge, it has been proposed by multiple authors. Using proxy outcome to do model selection is not novel either.

Writing:
I think the paper lacks a lot of the details, for example, what exactly is the novel stacking algorithm, what is the definition of AllMeta and what meta learners are included in this 'allmeta' definition. The value of the paper comes from the experiments and without very specific details it is hard to evaluate the quality.

**Questions:**

See above.

---

> ### Author Response · Authors · 2024-11-16
> **Response to reviewer mUTg [Part 1/3]**
>
> Thank you for reviewing our work! We respond to each point below: novelty and significance of our contributions, stacking procedure, and writing.
> ___
>
> ## Novelty and significance
>
> First, we would like to clarify the problem we address and more clearly describe the unique contributions of our work.
>
> **Problem statement**
>
> Recent years have seen significant advances in using ML for CATE estimation. However, many questions remain regarding the application of these methods in practice, which hampers their real-world adoption.
> First, many methods have been proposed for CATE estimation, but we do not yet clearly enough understand _when_ what methods are preferable. It even remains an open question how to best tune these methods, as no established guidelines exist. Besides modelling, aspects such as preprocessing or ensembling have received almost no consideration in previous work.
> Second, while there has been recent progress on understanding model selection in this context, different studies use vastly different setups and come to different conclusions–meaning that there is no consensus as to which approach works best.
>
> **Our solution**
>
> We argue that a _different perspective_ is needed to tackle these questions and bridge the gap between theoretical advancements in CATE estimation and their real-world application. We focus on automatically building and finding an optimal ML pipeline for CATE estimation: our proposed solution, AutoCATE, is the first approach that can tackle this end-to-end and in an automated manner. Both these characteristics are _important_ and _novel_, representing significant advances.
> Adopting an **end-to-end** approach enables us to consider _all relevant design choices_–such as preprocessing, model selection, and ensembling–together, rather than in isolation.
> Our emphasis on **automation** ensures the replicability and practicality of our protocols. This way, our final configuration of AutoCATE is a **single algorithm** for finding an optimal ML pipeline that gives state-of-the-art predictions across a variety of settings.
>
> **Methodological contributions**
>
> Developing this solution requires overcoming **significant challenges**. While individual components of our approach appear in prior work, a core contribution of AutoCATE lies in integrating these components into a fully automated, end-to-end framework. For example, while previous work focuses on comparing risk measures for model selection, we additionally analyze how to construct the ML pipelines underlying these risk measures. This is a clear consideration in practice, which had not yet been tackled in prior work. Finally, through this automated, end-to-end perspective, we uncover novel solutions to critical issues, such as how to effectively split data for training and validation or how to perform multi-objective optimization with multiple risk measures.
>
> As such, AutoCATE makes **several distinct and important contributions**. Methodologically, we provide the first unified algorithm that automates all stages of CATE estimation, from preprocessing to estimation, model selection and ensembling. Some functionalities of our framework are even _novel methodologically_, such as the multi-objective optimization, or re-discover methods that have been largely abandoned, such as the Lo-Learner.

---

> > ### Author Response · Authors · 2024-11-16
> > **Response to reviewer mUTg [Part 2/3]**
> >
> > **Empirical contributions**
> >
> > In addition to its methodological contributions, AutoCATE makes significant *empirical contributions*. Prior studies examine questions related to one of the three stages considered in AutoCATE in isolation. Conversely, our more holistic experimental setup provides more practical conclusions that lead directly to implementable solutions within our AutoCATE framework, which marks a substantial shift toward actionable guidance for both researchers and practitioners. This way, we can progress beyond earlier findings by revisiting similar questions in a more practical context enabled by automation. Our analyses uncover novel insights, challenging the established knowledge in CATE estimation. For example, Figure 1 shows that vastly different metalearners can be optimal for a given data generating process and highlights our limited understanding of this problem.
> >
> > Other fields have long benefited from meta-analyses and systematic comparisons, enabling more refined insights into model performance across scenarios. However, a similar level of synthesis and clarity is currently _missing_ in ML for CATE estimation, underscoring a critical need for comprehensive, end-to-end solutions that address these challenges. Through our framework, we conduct arguably the **most comprehensive analysis to date** of methods for both CATE estimation and model selection. We evaluate a wide array of data sets spanning classification and regression outcomes, varying in size, dimensionality, and data-generating processes. Similarly, we investigate a vastly more diverse collection of methods for CATE estimation and model selection, exploring their interplay in an automated and standardized framework. We emphasize that "benchmarks" are clearly included in ICLR’s call for papers.
> >
> > **Engineering contribution**
> >
> > A final contribution of our work is making AutoCATE accessible as an open-source software package. Our framework allows researchers to compare methods more easily, more fairly (based on similar tuning procedures), and more holistically (e.g., by analyzing whether a novel method is complementary), thereby supporting future innovations in this field. The low-code API will democratize access to these advanced methods by ensuring accessibility for practitioners unfamiliar with machine learning. Although our work is not limited to this package, we would like to emphasize that software can constitute an important research contribution. "Software libraries" is explicitly included as a topic in ICLR’s call for papers and many recent ICLR publications offer such contributions (e.g., Schneider, Balles & Hennig, 2019; Dangel, Kunstner & Hennig, 2020; Jarrett et al. 2021; Jiminez et al., 2024; Hvarfner, Hutter & Nardi, 2024).
> >
> > - **Action taken:** We have revised our paper to more clearly stress the addressed research gaps with respect to prior work and describe the contributions of our work.
> >
> > ## Stacking procedure
> >
> > As far as we are aware, the only other stacking procedure specifically tailored for CATE estimation is the one in Mahajan et al. (2023). While their approach finds weights $w_i = exp(\kappa M(\tilde{\tau}_i)$ by tuning a parameter $\kappa$, our alternative procedure solves a constrained optimization problem to find optimal weights $w_i$ with $0 \leq w_i \leq 1$.
> >
> > We agree that their approach constitutes a viable alternative approach. Therefore, we have included it as an option in our framework. We have also extended the procedure proposed by Mahajan et al. to also accommodate multiple objectives, ensuring that it is fully compatible with our framework. To provide additional clarity, we have added a more detailed description of all model selection and ensembling techniques in the appendix of the revised paper.
> >
> > We have added empirical results comparing our stacking procedure (COP, based on a constrained optimization problem) and theirs (Softmax). These results have been included in Table 3 of the updated paper. For your convenience, we compare their results (for a single T-Risk) below:
> >
> > ### Comparing approaches for stacking
> > |         | **COP**         | **Softmax**    |
> > |--------:|:---------------:|:--------------:|
> > | _IHDP_  | 1.96 $\pm$ .34  | 2.83 $\pm$ .51 |
> > | _ACIC_  | 1.42 $\pm$ .09  | 1.33 $\pm$ .09 |
> > | _Twins_ | .344 $\pm$ .00  | .331 $\pm$ .00 |
> > | _News_  | 2.33 $\pm$ .06 | 2.33 $\pm$ .06 |
> >
> > Although we are not aware of other methods for ensembling tailored for CATE estimation, we would be happy to include them if the reviewer can point them out to us. Unfortunately, conventional approaches for stacking can unfortunately not be applied for CATE, as we do not have a ground truth. We hope that our work will inspire researchers to develop novel methods for ensembling in this context.
> >
> > - **Action taken:**
> > We have included a more detailed explanation of our ensembling methods in the appendix.
> > We have included the procedure proposed in Mahajan et al. (2023) as part of our framework and our experiments.

---

> > > ### Author Response · Authors · 2024-11-16
> > > **Response to reviewer mUTg [Part 3/3]**
> > >
> > > ## Writing
> > > Thank you for highlighting areas where additional detail would enhance the clarity and value of our work. We have revised our writing throughout the paper again to include sufficient detail for evaluating the quality of the experiments.
> > >
> > > - **Action taken:** We have carefully addressed each of these points in the revised version:
> > >    - We clarified the explanation of our novel stacking algorithm and provided more detailed information in the appendix.
> > >    - In the AllMeta configuration, AutoCATE automatically searches _across_ all metalearners included in our search space: S, T, DR, X, R, RA, Lo, Z, U, and F. To the best of our knowledge, these comprise all metalearners previously proposed in the literature. We have formulated this more clearly in the revised version.
> > >    - We have added more detail explaining the captions of the experiments, where we point out which configuration of AutoCATE is used.
> > >
> > > If there are any remaining areas where additional clarification would be beneficial, we would be happy to address those as well.
> > > ___
> > >
> > > Once again, we would like to thank the reviewer for their time and effort in reviewing our work. While we hope that these clarifications and additions address your concerns, we would be happy to engage further if needed.
> > >
> > > ___
> > >
> > > ### References:
> > > - Schneider, F., Balles, L., & Hennig, P. DeepOBS: A Deep Learning Optimizer Benchmark Suite. (2019). In International Conference on Learning Representations.
> > > - Dangel, F., Kunstner, F., & Hennig, P. (2020). BackPACK: Packing more into Backprop. In International Conference on Learning Representations.
> > > - Jarrett, D., Yoon, J., Bica, I., Qian, Z., Ercole, A., & van der Schaar, M. (2021). Clairvoyance: A Pipeline Toolkit for Medical Time Series. In International Conference on Learning Representations.
> > > - Jimenez, Carlos E., John Yang, Alexander Wettig, Shunyu Yao, Kexin Pei, Ofir Press, and Karthik R. Narasimhan. (2024). "SWE-bench: Can Language Models Resolve Real-world Github Issues?." In The Twelfth International Conference on Learning Representations.
> > > - Kahl, Kim-Celine, Carsten T. Lüth, Maximilian Zenk, Klaus Maier-Hein, and Paul F. Jaeger. (2024). "ValUES: A Framework for Systematic Validation of Uncertainty Estimation in Semantic Segmentation." In The Twelfth International Conference on Learning Representations.
> > > - Hvarfner, C., Hutter, F., & Nardi, L. (2024). A General Framework for User-Guided Bayesian Optimization. In The Twelfth International Conference on Learning Representations.

---

> > > > ### Author Response · Authors · 2024-11-24
> > > >
> > > > Dear Reviewer,
> > > >
> > > > Thank you again for the time and effort you’ve dedicated to reviewing our submission!
> > > >
> > > > As the rebuttal period is drawing to a close, we wanted to kindly check if our responses have addressed your concerns. If so, we would greatly appreciate it if you could consider updating your score accordingly.
> > > >
> > > > Of course, we remain happy to engage further or clarify any additional questions or comments you may have.
> > > >
> > > > Thank you again for your thoughtful feedback and support during this process!
> > > >
> > > > Best regards,
> > > > Authors of Submission 3712

---

> > > > > ### Author Response · Authors · 2024-11-27
> > > > >
> > > > > Dear Reviewer,
> > > > >
> > > > > As the rebuttal period is nearing its end, we wanted to kindly follow up to see if our responses have addressed your concerns. If they have, we would be grateful if you could consider updating your score accordingly.
> > > > >
> > > > > If there are any remaining questions or points you would like us to address, please do not hesitate to let us know. We are more than happy to provide further clarifications or engage in additional discussion.
> > > > >
> > > > > Thank you once again for your time and thoughtful feedback throughout this process!
> > > > >
> > > > > Best regards,
> > > > > Authors of Submission 3712

---

> > > > > > ### Author Response · Authors · 2024-11-29
> > > > > >
> > > > > > Dear Reviewer,
> > > > > >
> > > > > > We hope you are doing well! With the rebuttal period extension, we wanted to kindly check in to see if our responses have addressed your concerns. If so, we would be incredibly grateful if you could consider updating your score, as two other reviewers have already done.
> > > > > >
> > > > > > If there are any lingering questions or areas where further clarification is needed, please let us know, and we would be more than happy to address them promptly.
> > > > > >
> > > > > > Thank you again for your thoughtful feedback! Wishing you a very Happy Thanksgiving!
> > > > > >
> > > > > > Best regards,
> > > > > >
> > > > > > The Authors of Submission 3712

---

> ### Author Response · Authors · 2024-12-02
>
> Dear Reviewer,
>
> As the rebuttal period is now coming to an end, we wanted to send a final reminder to kindly ask if our responses have addressed your concerns. If so, we would be deeply grateful if you could consider updating your score, as three other reviewers have now already done.
>
> Thank you again for your effort and thoughtful feedback.
>
> Best regards,
>
> The Authors of Submission 3712

---

### Meta-Review · Area_Chair_ud5g · 2024-12-23

**Metareview:**

**Summary**: The paper proposes AutoCATE an AUTOML analogous framework for CATE estimation, for scalable observational causal inference. The main distinction compared to prior work seems to be changes to the framework and the focus on a discussion of model selection of CATE nuisance parameters, selection of choice of CATE estimators, and hyperparameter selection for generalizability. Authors benchmark on standard datasets, and open source their framework for broader consumption.

**Strengths**: The problem of model selection, estimator selection remains crucial in observational causal inference with little consensus on systematic frameworks for scalable inference. The framework completes existing analogous frameworks and attempts to improve some of the shortcomings of prior work. The experimentation is exhaustive and the overall paper is well written.

**Weaknesses**: Reviewers raised concerns about insufficient distinction from prior work in the initial version. In addition concerns were raised about the novelty of the methods used because the main contribution is more the framework as opposed to the  estimators themselves.

**Justification**: The authors have rebutted extensively to all initial concerns, including making clear the novelty of the contributions, and explaining the significance of the framework. Authors also conducted many additional experiments based on concerns raised, especially by Reviewer riY4. I have gone over the complete rebuttal and the reviews as well as reviewer comments. Overall there is an acknowledgement of the strong engineering contribution and the significance of the problem, and any lingering concerns have just come down to what constitutes as an "innovation" because AutoCATE effectively benchmarks existing estimators in a systematic manner and proposes new and much needed systematic frameworks for observational causal inference. Given the final lack of consensus, I will recommend a reject and recommend that authors to continue to improve the work for a future submission.

**Additional Comments On Reviewer Discussion:**

During discussion, reviewer riY4 was in favor of acceptance while authors, acknowledging the significance of the engineering contributions, and the open source framework that the community could build on, primarily pointed out at the lack of methodological innovation. While all other reviewers were in favor of acceptance if majority was in favor of acceptance, considering the importance of such a framework, in the end, there was no consensus sufficient to warrant acceptance. In going with reviewer consensus, I have recommended a reject.

---

### Decision · Program_Chairs · 2025-01-22

Reject